# Mixture of Scope Experts at Test: Generalizing Deeper Graph Neural Networks with Shallow Variants

**Gangda Deng**
University of Southern California
Los Angeles, USA
gangdade@usc.edu

**Hongkuan Zhou**
University of Southern California
Los Angeles, USA
hongkuaz@usc.edu

**Rajgopal Kannan**
DEVCOM ARL Army Research Office
Los Angeles, USA
rajgopal.kannan.civ@army.mil

**Viktor Prasanna**
University of Southern California
Los Angeles, USA
prasanna@usc.edu

## Abstract

Heterophilous graphs, where dissimilar nodes tend to connect, pose a challenge for graph neural networks (GNNs). Increasing the GNN depth can expand the scope (*i.e.*, receptive field), potentially finding homophily from the higher-order neighborhoods. However, GNNs suffer from performance degradation as depth increases. Despite having better expressivity, state-of-the-art deeper GNNs achieve only marginal improvements compared to their shallow variants. Through theoretical and empirical analysis, we systematically demonstrate a shift in GNN generalization preferences across nodes with different homophily levels as depth increases. This creates a disparity in generalization patterns between GNN models with varying depth. Based on these findings, we propose to improve deeper GNN generalization while maintaining high expressivity by Mixture of scope experts at test (`Moscat`). Experimental results show that `Moscat` works flexibly with various GNNs across a wide range of datasets while significantly improving accuracy. Our code is available at https://github.com/Hydrapse/moscat.

## 1 Introduction

Graph neural networks (GNNs) are emerging as powerful tools for graph mining applications, such as social recommendations [17], traffic prediction [27], and fraud detection [49]. GNNs' superior performance is largely attributed to graph *homophily*, where similar nodes tend to be connected. However, some graphs exhibit *heterophily*, where connected nodes tend to be dissimilar. When aggregating heterophilous information, GNNs tend to generate similar embeddings for nodes of different classes, leading to suboptimal performance.

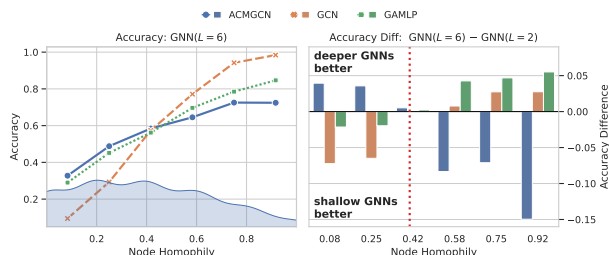

Figure 1: Performance on `amazon-ratings`. (Left) Deeper GNNs exhibit performance disparities across node subgroups with different homophily ratios, with the shaded area indicating the node distribution. (Right) Deeper GNNs and their shallow variants show a shift in generalization preference across homophily ratios, with the red dotted line indicating the average training set homophily (0.38).

39th Conference on Neural Information Processing Systems (NeurIPS 2025).

To find homophily on graphs where heterophily dominates, GNNs need to search for neighbors from higher hops. Typically, a GNN with $L$ layers of propagation can perceive the entire $L$-hop neighborhood for each node, making the *scope* (i.e., receptive field) size tightly coupled with the GNN's depth. However, a persistent challenge is that GNNs experience performance degradation as they become deeper. Studies have shown that deeper GNNs suffer from three key issues: reduced expressivity due to over-smoothing [32, 72], model degradation from optimization challenges [80, 41], and large generalization gaps caused by overfitting [73]. While many techniques have been proposed to address the first two issues, these solutions often sacrifice generalization [13], resulting in limited overall performance improvements.

Existing theoretical analyses of deeper GNN generalization adopt a purely global perspective [13, 64], which overlooks the diversity of local structural patterns (e.g., homophily) that appear in real-world graphs. For example, Figure 1 (left) shows that variations in homophily can induce substantial performance discrepancies for deeper GNNs. Recent work has moved toward more realistic assumptions by modeling graphs as mixtures of node subgroups with varying homophily levels [45, 39]. However, these theories are restricted to one- or two-layer GNNs and thus fail to reveal the generalization behavior of GNNs with increasing depths.

To address this gap, we derive a new PAC-Bayesian generalization bound, which reveals a *shift* in model generalization preferences across nodes with different homophily levels as depth increases. This provides a new understanding of deeper GNNs' failure: while deeper GNNs achieve better generalization than their shallow variants in either homophily or heterophily region, they sacrifice generalization performance in the other region, thus limiting overall performance. Figure 1 (right) demonstrates the generalization preference shift with increasing depth in real-world data.

These insights motivate us to learn the generalization patterns of GNN experts with varying depths with a gating model, which can combine the advantages of both shallow and deeper GNNs. We first examine existing approaches to handle GNN depth, which can be broadly categorized into two paradigms: "Deeper GNN" and "Graph MoE" (Figure 2 left). "Deeper GNN" typically uses skip connections to mix embeddings from different scopes, while "Graph MoE" stacks varying numbers of GNN layers as experts for their corresponding scopes. These methods use gating mechanisms

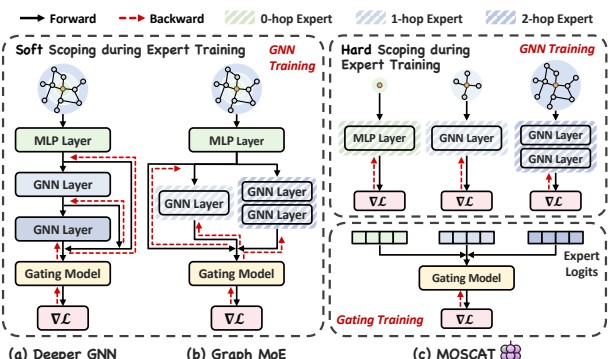

Figure 2: The landscape of GNNs with scope mixing.

to mix knowledge from different scope experts and impose *soft scoping*: while each expert encodes only its own scope during forward propagation, gradients from higher-hop information still flow back to the shallow experts through the gating model. Although this can be expressive, shallow experts tend to overfit to noise from higher-hop neighbors, resulting in suboptimal performance. For empirical evidence, please refer to Appendix G.5.

To address this issue, we propose a new paradigm that decouples the gating module from GNN training. As illustrated in Figure 2 (right), we first train each expert independently. Then, a specialized gating model learns to node-adaptively combine predictions from shallow experts to enhance the generalization of the deeper expert. This enforces *hard scoping*: each expert can only learn from the knowledge available in its specific scope. Notably, the gating model should be trained on a holdout set to accurately measure the generalization preference of experts.

Based on this paradigm, we develop a concrete method named `Moscat`: Mixture of scope experts at test—a post-processing attention-based gating model that seamlessly integrates with various GNN architectures to exploit depth. We further develop two techniques: heterophily-biased sample filtering and scope-aware logit augmentation to address technical challenges in scope expert mixing (detailed in Section 4). Our main contributions are summarized as follows:

**A novel paradigm:** We propose a novel decoupled expert-gating paradigm that learns generalization patterns of GNNs with different depths, supported by our theoretical analysis.

**Superior flexibility:** `Moscat` detects various patterns of expert failure, enabling better gating that significantly improves performance across various GNN architectures.

**SOTA performance:** `Moscat` outperforms state-of-the-art GNNs, Graph Transformers, and Graph MoEs across a wide range of datasets with varying sizes and homophily ratios.

## 2   Preliminaries

We denote a graph as $\mathcal{G}\left(\mathcal{V}, \mathcal{E}\right)$ with node set $\mathcal{V} = \{v_i\}_{i=1}^n$, edge set $\mathcal{E} \subseteq \mathcal{V} \times \mathcal{V}$. Nodes are associate with node feature matrix $\mathbf{X} \in \mathbb{R}^{|\mathcal{V}| \times F}$ and one-hot class label matrix $\mathbf{Y} \in \mathbb{R}^{|\mathcal{V}| \times C}$. By $\mathcal{N}_v$, we denote the neighbors of $v$, which is the set of nodes directly connected to $v$. Let $\mathbf{A}$ be the adjacency matrix, $\mathbf{D}$ be the diagonal degree matrix and $\widetilde{\mathbf{A}} = \widehat{\mathbf{D}}^{-\frac{1}{2}} \widehat{\mathbf{A}} \widehat{\mathbf{D}}^{-\frac{1}{2}}$ be the normalized adjacency matrix, where $\widehat{\mathbf{A}} = \mathbf{A} + \mathbf{I}$, $\widehat{\mathbf{D}} = \mathbf{D} + \mathbf{I}$, and $\mathbf{I}$ is the identity matrix. Further preliminary details and related work are in Appendix C and A.

**Node Homophily.** Homophily metrics are widely used as graph properties to measure the probability of nodes with the same class connected to each other. In this paper, we mainly use a node-wise homophily metric called *node homophily* [52]. It defines the fraction of neighbors that have the same class for each node $v$: $|\{u \in \mathcal{N}_v : y_u = y_v\}| / |\mathcal{N}_v|$.

**Node subgroup.** The nodes in a graph can be divided into non-overlapping node subgroups, with each subgroup containing nodes that share similar properties (e.g., node homophily).

## 3   Understanding Generalization Disparity across GNN Scope Experts

### 3.1   Unpacking the Depth Dilemma: Why Do GNNs Struggle with Generalization?

Existing studies have proved that deeper GNNs are more expressive [50, 25, 48, 10]. In practice, however, performance degradation is widely observed when going deep. Unlike previous studies that attribute the failure of deeper GNNs to a single cause, we argue that this failure stems from multiple factors, varying based on GNN architectures. To analyze each factor separately, we break down the error of deeper GNNs on unseen samples. Let $\mathcal{F}$ denote the function class for a given deeper GNN architecture, and $f_S \in \mathcal{F}$ denote a function learned on the training set $S$. The population risk (true risk) of $f_S$ over the entire data distribution can be decomposed as:

$$R\left(f_S\right) = \underbrace{\hat{R}_S\left(f_{S,ERM}\right)}_{\text{representation error}} + \underbrace{\hat{R}_S\left(f_S\right) - \hat{R}_S\left(f_{S,ERM}\right)}_{\text{optimization error}} + \underbrace{R\left(f_S\right) - \hat{R}_S\left(f_S\right)}_{\text{generalization error}}, \tag{1}$$

where $\hat{R}_S$ is the empirical risk (training error) and $f_{S,ERM}$ is the empirical risk minimizer. In the following, we discuss these errors and connect them with recent findings: **(1)** *Representation error* measures the minimum error over the function class $\mathcal{F}$ on $S$, evaluating the expressive power of the GNN architecture. A major factor limiting deeper GNNs' expressivity is *over-smoothing* [32], where node representations become indistinguishable after multiple GNN layers. Recent studies [80] show that GNNs without non-linearity between propagation layers (e.g., SGC) tend to suffer from over-smoothing. **(2)** *Optimization error* depicts how well $f_S$ trained by calculating the risk difference between the learned model $f_S$ with the empirical risk minimizer $f_{S,ERM}$. Research shows that deeper GNNs encounter significant training difficulties [41] and exhibit *model degradation* [80] as depth increases, leading to decreased accuracy in both training and testing. The primary cause is gradient-related issues such as vanishing gradients [30] and gradient instability [13]. These problems can be mitigated through optimization tricks like skip-connections. **(3)** *Generalization error* measures how well the trained model $f_S$ generalizes from the training set $S$ to the true distribution. The *overfitting* [73, 13] issue has been identified to explain why well-trained deeper GNNs with higher expressivity failed to outperform their shallow variants. To address this issue, existing works typically reduce GNN parameters or employ regularization techniques.

While useful, existing solutions face inherent trading-offs between these three types of errors (See Appendix D for detailed analysis). Given these theoretical limitations and the ineffective use of depth in practice, a crucial question emerges: *Is it possible to enhance the generalization capabilities of deeper GNNs without compromising their expressivity or increasing training difficulty?*

## 3.2 Subgroup Generalization Bound for GNNs with Varying Scopes: A Data-Centric Perspective

To investigate this question, we conduct a theoretical analysis of how GNNs generalize at different depths. Rather than focusing on model architecture [13], we take a data-centric perspective to understand generalization across diverse local structural patterns. Recent work [45, 39] has shown that real-world graphs comprise node subgroups with varying homophily levels, causing GNNs to exhibit generalization discrepancies across these subgroups. However, these studies only focus on simplified GNNs with one or two layers, which fails to explain deeper GNNs' failure. Though deeper GNNs have superior expressivity and may generalize better on specific subgroups, these improvements often get obscured by overall performance degradation. To rigorously analyze generalization disparity with respect to GNN depth and examine the role of homophily, we derive a new generalization bound for multi-layer GNNs by extending [45]'s non-i.i.d. PAC-Bayesian analysis on GNNs with one-hop aggregation. Following the assumptions used in [45], we adopt the contextual stochastic block model (CSBM) with 2 classes for a controlled study, which is widely used for graph analysis [4, 67].

**Assumption 3.1** (CSBM-Subgroup dataset). The generated nodes consist of two disjoint sets $\mathcal{C}_1$ and $\mathcal{C}_2$ of the same size. The features for nodes belonging to $\mathcal{C}_1$ and $\mathcal{C}_2$ are sampled from $N(\boldsymbol{\mu}_1, \mathbf{I})$ and $N(\boldsymbol{\mu}_2, \mathbf{I})$, respectively. Each set consists of $M$ subgroups. Each subgroup $m$, appears with probability $\Pr(m)$, has probabilities of intra-class edge $p^{(m)}$ and inter-class edge $q^{(m)} = 1 - p^{(m)}$. The dataset can be denoted as $\mathcal{D}\big(\boldsymbol{\mu}_1, \boldsymbol{\mu}_2, \{(p_m, q_m); \Pr(m)\}_{m=1}^M\big)$.

Since the key distinction between deeper neural networks and deeper GNNs is the expansion of scopes, we use SGC as our GNN model. SGC decouples scope enlargement from the addition of linear transformations, allowing us to specifically analyze how varying scopes impact generalization.

**Assumption 3.2** (GNN model). We use the following architecture: $f^{L'}\big(g^L(\mathbf{X}, \mathcal{G}); \mathbf{W}^{(1)}, \mathbf{W}^{(2)}, \cdots, \mathbf{W}^{(L')}\big)$, where $g^L$ denotes an $L$-hop mean aggregation function and $f^{L'}$ is a ReLU-activated $L'$-layer MLP with hidden dimension $d$.

The following theorem is based on the PAC-Bayes analysis with margin loss [44, 12]. We aim to bound the generalization gap between the expected loss $\mathcal{L}_m^0(\theta)$ of a test subgroup $m$ for 0 margins and the empirical loss $\widehat{\mathcal{L}}_S^\gamma(\theta)$ on train subgroup $S$ for a margin $\gamma$.

**Theorem 3.3** (GNN Subgroup generalization bound). *Assume the aggregated features $g^L(\mathbf{X}, \mathcal{G})$ share the same variance $\sigma^2 \mathbf{I}$. Let $\theta$ be any classifier in the parameter set $\{\mathbf{W}^{(l)}\}_{l=1}^{L'}$ and $S$ denote the training set. For any test subgroup $m \in \{1, \cdots, M\}$ and large enough number of the training nodes $N_S = |\mathcal{V}_S|$, with probability at least $1 - \delta$ over the sample $\{y_v\}_{v \in V_S}$, there exists $0 < \alpha < \frac{1}{4}$ we have:*

$$\mathcal{L}_m^0(\theta) - \widehat{\mathcal{L}}_S^\gamma(\theta) \leq \mathcal{O}\Big(\frac{\rho}{\sigma^2}\Big(\epsilon_m + \rho\,(p_S - p_m)\,\Gamma_{L-1}\Big)\Big) + \mathcal{O}\Big(\frac{\|W\|^2\,(\epsilon_m)^{2/L'}}{N_S^\alpha}\Big) + \mathcal{O}\Big(\frac{\ln(1/\delta)}{N_S^{2\alpha}}\Big),$$

(2)

*where $\|W\|^2 := \sum_{l=1}^{L'} \|\widetilde{W}_l\|_F^2$, $\rho := \|\boldsymbol{\mu}_1 - \boldsymbol{\mu}_2\|$ is feature distribution separability, $\epsilon_m := \max_{u \in V_m} \min_{v \in V_S} \big\|g^L(\mathbf{X}, \mathcal{G})_u - g^L(\mathbf{X}, \mathcal{G})_v\big\|_2$ is the bound of the aggregated feature distance, and $\Gamma_{L-1} := \mathbb{E}_{o \sim \Pr(o), o \in \{1, \ldots, M\}} \Big[(p_o - q_o)^{L-1}\Big]$ represents $L$-hop homophily coefficient.*

Proofs and details are in Appendix B. With a sufficiently large training set $N_S$, the bound is dominated by the first term. It has the following properties. **Prop.1:** The generalization error for each subgroup $m$ depends on the aggregated feature distance $\epsilon_m$ as well as the homophily ratio difference $p_S - p_m$. **Prop.2:** Consider any two subgroups $i$ and $j$ where $p_i > p_S$ and $p_j < p_S$, respectively. Due to the decay of $|\Gamma_{L-1}|$ when increasing $L$, the minimum achievable generalization error occurs at different depth $L$ for these two subgroups. **Prop.3:** Varying $L$ yields a larger disparity in generalization error on heterophilous graphs (where $\Gamma_{L-1} \in [-1, 1]$) than on homophilous graphs (where $\Gamma_{L-1} \in [0, 1]$).

Theorem 3.3 suggests that varying the depth shifts the GNN generalization pattern across subgroups. This creates a notable disparity in subgroup generalization between shallow and deeper GNNs, especially on heterophilous graphs. In particular, the generalization disparity emerges between test subgroups with low and high homophily—partitioned by the average homophily ratio of training set.

### 3.3 Performance Disparity across Scope Experts on Real-World Datasets

In this subsection, we empirically examine our theoretical analysis from multiple perspectives. **First**, we use Jaccard Coefficient to calculate the overlapping ratio of correctly predicted test nodes between each pair of depths. Figure 3 (Top) shows that all overlapping ratios are relatively small, indicating that each variant can correctly predict a unique subset of nodes.

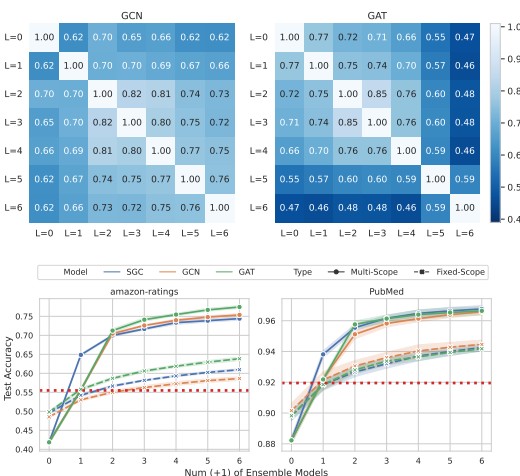

The ratio between deeper and shallow variants is even lower. On `Penn94`, GCN models with depth $L \geq 2$ deviates from its shallow variants, while GAT shows deviation at greater depths ($L \geq 5$). **Second**, we further compare the performance of deeper and shallow GNNs across node subgroups with different homophily levels. As shown in Figure 1 (Right), shallow and deeper GNN variants exhibit a significant generalization disparity between subgroups partitioned by the average homophily ratio of training set. **Third**, we examine the significance of the observed generalization disparity. Since training randomness can lead to performance variations, we compare the *union* of nodes correctly predicted by (1) models of different depths and (2) models with the same depth but different random seeds. Figure 3 (Bottom) demonstrates that the performance gains from increased depth significantly exceed those from training stochasticity, particularly on the heterophilous graph (`amazon-ratings`) compared to the homophilous graph (`PubMed`). We defer additional empirical evidence and analysis to Appendix G.9.

Figure 3: (Top) The overlapping ratio on `Penn94`. (Bottom) Test accuracy under *Oracle* ensemble. *Multi-Scope* represents the ensemble of GNNs with depths ranging from $L = 0$ (MLP) to $n$ ($n \leq 6$). *Fixed-Scope* represents the ensemble of GNNs with identical depth $L = L_{\text{best}}$. The horizontal red dotted line shows the SOTA GNN accuracy.

In summary, our theoretical and empirical analysis provides a new understanding of GNNs' depth dilemma: Increasing GNN depth enhances generalization for certain subgroups but inevitably compromises generalization for others, leading to suboptimal overall performance. This insight demonstrates the value of combining predictions from deeper and shallow GNN models during inference—an approach that improves overall generalization while maintaining expressivity and avoiding additional training complexity.

## 4 Proposed Method: `GNN-Moscat`

Moving from the paradigm (Figure 2 right) to practical implementation, several technical challenges emerge: **(1) Feature construction.** Although node homophily is a strong identifier for experts' generalization disparity (Section 3.2), it's not available during testing. **(2) Training sample selection.** The training set for experts is noisy for training the gating model since experts can all achieve high accuracy on the training set but cannot generalize equality well. **(3) Diverse expert architectures.** Expert failures may stem from complex reasons—such as over-smoothing, model degradation (underfitting), and overfitting (Section 3.1)—that vary across different expert architectures.

We introduce `Moscat`, a post-processing gating model for scope experts that successfully addresses these challenges. Figure 4 presents an overview of our proposed method. 4.1 presents the overall model and 4.2 discuss the properties.

### 4.1 The MoE Workflow

**Scope Experts Training.** Different depth GNN models serving as experts of their corresponding scope is a hot topic in recent Graph MoE research [61, 78, 75]. For decoupled GNNs like SGC, we consider the number of propagation layers as depth while keeping the transformation layers fixed. Formally, given a GNN architecture $\mathcal{M}$ and the maximum number of layers $L_{\max}$, we independently train $L_{\max} + 1$ models $\{\mathcal{M}^0, ..., \mathcal{M}^{L_{\max}}\}$ with depth from 0 to $L_{\max}$, where $\mathcal{M}^0$ is an MLP.

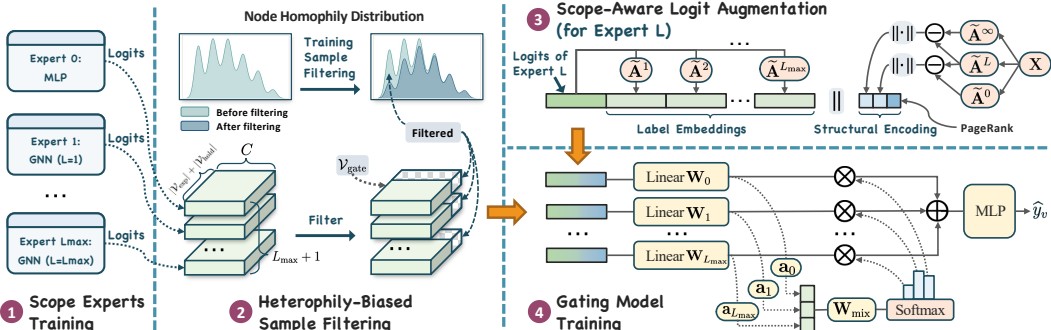

Figure 4: Overview of `Moscat`. (1) Different-depth GNN models serve as scope experts (with MLP as 0-hop), each trained independently. (2) Collect logits from each expert by running inference on the expert-training set $\mathcal{V}_{\text{exp}}$ and a holdout set $\mathcal{V}_{\text{hold}}$, then perform heterophily-biased filtering to form the gating-training set $\mathcal{V}_{\text{gate}}$. (3) Enhance logits with label embeddings and structural encoding. (4) Train the gating model using node labels, with learnable parameters shown in yellow blocks.

**Holdout Set for Gating Model.** Let $\mathcal{V}_{\text{exp}}$ denote the training set for the experts. To understand how experts generalize, we reserve a holdout set $\mathcal{V}_{\text{hold}}$ from the labeled data for training the gating model $\phi$, ensuring $\mathcal{V}_{\text{exp}} \cap \mathcal{V}_{\text{hold}} = \varnothing$. To expand the gating model's training data, we may optionally include a subset of $\mathcal{V}_{\text{exp}}$ to form the final gating training set $\mathcal{V}_{\text{gate}} \subseteq \mathcal{V}_{\text{exp}} \cup \mathcal{V}_{\text{hold}}$.

**Heterophily-Biased Sample Filtering.** However, samples in $\mathcal{V}_{\text{exp}}$ can be noisy for gating model $\phi$ training, which fall into two categories: (1) Samples that are *underfitted* or properly fitted by all experts are ideal for $\phi$ training, as they accurately reflect each expert's performance. (2) Samples that are overfitted by some experts become problematic for $\phi$ training since $\phi$ will mistakenly assign high weights to the overfitted experts for that subgroup. Importantly, we notice that experts tend to overfit to heterophilous nodes $\mathcal{V}_{\text{exp-het}}$. This is because heterophily nodes exhibit more diverse neighborhood label patterns than homophily nodes, making them more challenging for experts to generalize (See Appendix G.6). To address these cases, we introduce a hyperparameter $\gamma \in [0, 1]$ that randomly filters out samples in $\mathcal{V}_{\text{exp-het}}$. We also add a binary hyperparameter to determine whether to use $\mathcal{V}_{\text{exp}}$ for gating training. We denote the node sample dropped from $\mathcal{V}_{\text{exp}}$ as $\mathcal{V}_{\text{exp-drop}} \in \{\mathcal{V}_{\text{exp}}, \mathcal{V}_{\text{exp-het}}^{(\gamma)}\}$.

Additionally, certain nodes present a particular challenge where all experts fail to make correct predictions. These difficult samples $\mathcal{V}_{\text{all-wrong}} \subseteq \mathcal{V}_{\text{exp}} \cup \mathcal{V}_{\text{hold}}$, found primarily in the heterophily region, fall into two categories: (1) Cases where individual experts make incorrect predictions, but their logits still contain valuable structural patterns that contribute to task prediction. (2) Cases where all experts generate meaningless predictions, either due to inherent dataset noise or architectural limitations. To address both scenarios, we introduce a binary hyperparameter to control the masking of these samples $\mathcal{V}_{\text{wr-drop}} \in \{\mathcal{V}_{\text{all-wrong}}, \varnothing\}$. The final training set for the gating model $\phi$ is:

$$\mathcal{V}_{\text{gate}} := \left( \mathcal{V}_{\text{hold}} \cup (\mathcal{V}_{\text{exp}} \setminus \mathcal{V}_{\text{exp-drop}}) \right) \setminus \mathcal{V}_{\text{wr-drop}} \tag{3}$$

**Scope-Aware Logit Augmentation.** Expert logits serve as a crucial input for the gating model, encoding both structural information and the expert's confidence. However, experts may suffer from *overfitting*, showing strong confidence despite poor generalization. Theorem 3.3 shows that experts' generalization capability across subgroups can be identified by node homophily, which is defined as the similarity between a node's own label distribution and its neighborhood's label distribution. We extend this concept to higher-hop neighbors and approximate it using pseudo-label distribution. Let $\mathbf{Z}^{(L)}$ denote the logits for expert $\mathcal{M}^L$. We propose *label embeddings* to augment $\mathcal{M}^L$'s logits with pseudo neighborhood label distribution from 1 to $L_{\max}$ hops: $\boldsymbol{\xi}_{\text{label}}^{(L)} = \left[ \widetilde{\mathbf{A}}^1 \mathbf{Z}^{(L)} \| \cdots \| \widetilde{\mathbf{A}}^{L\max} \mathbf{Z}^{(L)} \right]$.

When an expert lacks expressivity, it can suffer from *over-smoothing* as depth increases. To help identify when over-smoothing happens in experts, we augment their logits with *structural encoding*, incorporating node smoothness and centrality. Let $\mathbf{X}^{(L)} = \widetilde{\mathbf{A}}^L \mathbf{X}$ represent node features smoothed within the scope size $L$. We measure its distance from both the original feature $\mathbf{X}^{(0)} = \mathbf{X}$ and the final smoothed feature $\mathbf{X}^{(\infty)} = \widetilde{\mathbf{A}}^\infty \mathbf{X}$, where $\widetilde{\mathbf{A}}_{u,v}^\infty = (d_u + 1)^{\frac{1}{2}} (d_v + 1)^{\frac{1}{2}} / (2|\mathcal{E}| + |\mathcal{V}|)$ [32]. For each node $v$, we calculate two distance scalars $\bar{\epsilon}_v^{(L)} = \|\mathbf{X}_v^{(L)} - \mathbf{X}_v^{(0)}\|_2$ and $\tilde{\epsilon}_v^{(L)} = \|\mathbf{X}_v^{(L)} - \mathbf{X}_v^{(\infty)}\|_2$

to measure smoothness. We also incorporate PageRank centrality $\pi_v$ to encode node position. The structural encoding can be represented as $\boldsymbol{\xi}_{\text{struc}}^{(L)} = \left[ \bar{\boldsymbol{\epsilon}}^{(L)} \parallel \tilde{\boldsymbol{\epsilon}}^{(L)} \parallel \boldsymbol{\pi} \right]$. Finally, we combine these components to form the overall augmented logits for expert $\mathcal{M}^L$ as $\boldsymbol{\zeta}^{(L)} = \left[ \mathbf{Z}^{(L)} \parallel \boldsymbol{\xi}_{\text{label}}^{(L)} \parallel \boldsymbol{\xi}_{\text{struc}}^{(L)} \right]$, where $\boldsymbol{\zeta}^{(L)} \in \mathbb{R}^{|\mathcal{V}| \times F_{\text{aug}}}$ and $F_{\text{aug}} = C + L_{\max} C + 3$.

**Gating Model Training.** Taking the augmented logits from all experts as input $\{\boldsymbol{\zeta}^{(0)}, ..., \boldsymbol{\zeta}^{(L_{\max})}\}$, we train the gating model $\phi$ on $\mathcal{V}_{\text{gate}}$, using node labels $\mathbf{Y}$. Figure 4-4 illustrates the architecture of our gating model $\phi$. Moscat uses an attention-based gating mechanism to calculate weight $\boldsymbol{g}_{L,v}$ for expert $L$ on node $v$. $F_{\text{hid}}$ denotes the hidden dimension and $\mathbf{W}_L \in \mathbb{R}^{F_{\text{aug}} \times F_{\text{hid}}}, \mathbf{a}_L \in \mathbb{R}^{F_{\text{hid}}}, \mathbf{W}_{\text{mix}} \in \mathbb{R}^{(L_{\max}+1) \times (L_{\max}+1)}$ are learnable weights. Notably, it's crucial to separate transformation weights $\mathbf{W}_L$ and $\mathbf{a}_L$ for each expert. Since experts may suffer from over-smoothing and overfitting to varying degrees, each expert favors a specific combination of predefined structural patterns in $\boldsymbol{\zeta}^{(L)}$. Formally,

$$\hat{\boldsymbol{g}}_L = \texttt{Sigmoid}\left(\mathbf{H}_L \mathbf{a}_L\right), \ \mathbf{H}_L = \texttt{ReLU}\left(\boldsymbol{\zeta}^{(L)} \mathbf{W}_L\right), \ L \in \{0, ..., L_{\max}\},$$
$$[\boldsymbol{g}_0; \cdots ; \boldsymbol{g}_{L_{\max}}] = \texttt{Softmax}\left([\hat{\boldsymbol{g}}_0; \cdots ; \hat{\boldsymbol{g}}_{L_{\max}}] \mathbf{W}_{\text{mix}}\right), \tag{4}$$

$\boldsymbol{g}_L \in \mathbb{R}^{|\mathcal{V}|}$ is the node-adaptive gating weights for expert $L$. Then, we use a classifier $f(\cdot)$ to combine the knowledge from all experts to make predictions $\widetilde{\mathbf{Y}} = f\left(\sum_{L=0}^{L_{\max}} \texttt{diag}\left(\boldsymbol{g}_L\right) \mathbf{H}_L\right)$. Empirically, we find out that using a simple MLP as $f(\cdot)$ performs best compared to more complex models like GNNs or Transformers. For the gating model optimization, we employ the same loss function used to train the experts In this paper, we focus on the node classification task, therefore, we adopt the cross entropy loss (CE) to train the gating model: $\mathcal{L} = \mathbb{E}_{v \sim \widetilde{\mathcal{V}}_{\text{train}}} \texttt{CE}\left(\phi(\{\boldsymbol{\zeta}_v^{(L)}\}_{L=0}^{L_{\max}}), \mathbf{Y}_v\right)$.

### 4.2 Discussion and Analysis

**Runtime and space analysis.** Moscat is a lightweight module with relatively small training time and memory consumption. Please refer to Appendix G.2 and G.3 for detailed analysis.

**Comparison with LLM MoEs.** Unlike LLM MoEs with joint training, Moscat employs a separate "train-then-merge" strategy for GNN experts. Please refer to Appendix E for details.

**Comparison with ensemble methods.** Ensemble methods typically require extensive tuning of models with varying hyperparameters and architectures. In contrast, Moscat focuses on a controlled study of scope/depth effects by using identical architecture and hyperparameters across all experts.

## 5 Experiments

In this section, we aim to answer the following questions to verify the effectiveness of Moscat. **Q1**: How does Moscat perform in improving GNNs with varying architectures? **Q2**: How does Moscat compare to other techniques in enhancing deeper GNNs? **Q3**: How does each component of Moscat contribute to its overall performance? To understand why Moscat is effective, we further conduct a case study at the end of this section and defer other in-depth experimental analyses to the Appendix: Appendix G.7 investigates how expert training affects Moscat performance, Appendix G.8 provides a visual interpretation of how Moscat enhances GNN generalization, and Appendix G.10 analyzes AH's learned gating weights of different scope experts.

### 5.1 Experimental setup

**Datasets.** We evaluate Moscat on real-world datasets with varying node homophily ratios and graph sizes. We exclude commonly used small heterophilous graphs (fewer than thousands of nodes) due to their high variance and lack of statistical significance [53]. Please see Appendix F.1 for details.

**Evaluation Setup.** We assess model performance on node classification task using test accuracy (ROC-AUC on genius). We adopt two sample splits for our method: **(1) Moscat∗.** We split the training set into two disjoint subsets—one for training the GNN experts $\mathcal{V}_{\text{exp}}$ and another for the holdout set $\mathcal{V}_{\text{hold}}$. The split ratio serves as a tunable hyperparameter, and we do not use the original validation set for expert/gate training. **(2) Moscat.** Following prior work [29, 62], we leverage the

Table 1: `Moscat` exhibits large improvements over various base GNN architectures. Each ∗-`Moscat` (∗) combines predictions from 7 base GNN models with 0 (MLP) to 6 convolution layers. We report the best accuracy among these base GNN models as baselines.

| | Squirrel | snap-patents | arxiv-year | Flickr | amazon-ratings | Penn94 | genius | ogbn-arxiv | Avg.% |
|---|---|---|---|---|---|---|---|---|---|
| #Nodes | 2223 | 2.9M | 0.16M | 89250 | 24492 | 40000 | 0.4M | 0.16M | Improv. |
| #Edges | 46998 | 13M | 1.2M | 0.89M | 93050 | 1.3M | 1M | 2.3M | |
| Node Homo. | 0.16 | 0.19 | 0.28 | 0.32 | 0.38 | 0.48 | 0.51 | 0.64 | |
| MLP | 38.57 ±1.99 | 31.13 ±0.07 | 37.25 ±0.30 | 47.48 ±0.09 | 41.85 ±0.77 | 74.63 ±0.39 | 86.80 ±0.07 | 55.68 ±0.22 | |
| SGC | 40.04 ±1.77 | 48.71 ±0.10 | 45.88 ±0.32 | 52.07 ±0.15 | 49.58 ±0.55 | 81.17 ±0.40 | 88.01 ±0.20 | 71.89 ±0.10 | |
| SGC-Moscat∗ | 42.73 ±2.06 | 55.45 ±0.07 | 52.09 ±0.27 | 53.78 ±0.07 | 51.82 ±0.29 | 84.75 ±0.41 | 92.31 ±0.06 | 73.31 ±0.10 | ⇑ **6.05%** |
| SGC-Moscat | 43.35 ±1.97 | 55.16 ±0.06 | 51.86 ±0.30 | 54.40 ±0.11 | 52.72 ±0.53 | 84.80 ±0.49 | 92.29 ±0.09 | 73.94 ±0.22 | ⇑ **6.53%** |
| GCN | 41.33 ±1.46 | 50.79 ±0.16 | 48.68 ±0.34 | 55.52 ±0.36 | 48.55 ±0.38 | 82.54 ±0.43 | 90.22 ±0.26 | 72.13 ±0.17 | |
| GCN-Moscat∗ | 42.91 ±1.96 | 55.29 ±0.09 | 52.58 ±0.15 | 57.53 ±0.07 | 51.02 ±0.50 | 85.51 ±0.42 | 92.37 ±0.06 | 73.37 ±0.14 | ⇑ **4.25%** |
| GCN-Moscat | 43.55 ±2.08 | 55.29 ±0.07 | 53.00 ±0.18 | 57.75 ±0.16 | 52.25 ±0.69 | 85.76 ±0.32 | 92.37 ±0.06 | 74.09 ±0.32 | ⇑ **4.96%** |
| GAT | 39.36 ±1.89 | 44.45 ±0.33 | 52.77 ±0.32 | 55.87 ±0.28 | 49.78 ±0.47 | 81.81 ±0.62 | 88.44 ±1.06 | 72.01 ±0.22 | |
| GAT-Moscat∗ | 43.10 ±1.98 | 54.60 ±0.11 | 55.53 ±0.15 | 58.04 ±0.13 | 52.56 ±0.40 | 85.41 ±0.48 | 92.18 ±0.30 | 73.53 ±0.07 | ⇑ **6.29%** |
| GAT-Moscat | 43.10 ±2.13 | 54.77 ±0.09 | 56.06 ±0.28 | 58.52 ±0.19 | 53.77 ±0.61 | 85.35 ±0.59 | 92.21 ±0.31 | 74.13 ±0.17 | ⇑ **6.90%** |
| GCNII | 42.48 ±1.86 | 49.18 ±0.23 | 51.75 ±0.36 | 56.31 ±0.27 | 52.45 ±0.57 | 82.34 ±0.51 | 90.41 ±0.35 | 72.74 ±0.17 | |
| GCNII-Moscat∗ | 43.61 ±2.18 | 54.86 ±0.13 | 54.60 ±0.24 | 57.45 ±0.21 | 53.99 ±0.19 | 85.33 ±0.36 | 92.36 ±0.07 | 73.33 ±0.09 | ⇑ **3.59%** |
| GCNII-Moscat | 43.71 ±1.88 | 54.55 ±0.16 | 55.10 ±0.45 | 57.95 ±0.16 | 54.38 ±0.59 | 85.66 ±0.50 | 92.35 ±0.08 | 74.13 ±0.27 | ⇑ **4.05%** |
| ACMGCN | 35.00 ±2.56 | 48.90 ±0.15 | 45.45 ±0.37 | 54.51 ±0.34 | 53.37 ±0.42 | 83.38 ±0.47 | 64.74 ±5.55 | 71.85 ±0.41 | |
| ACMGCN-Moscat∗ | 42.81 ±2.09 | 55.04 ±0.08 | 50.72 ±0.22 | 56.40 ±0.24 | 56.02 ±0.50 | 85.79 ±0.23 | 91.90 ±0.14 | 73.23 ±0.07 | ⇑ **11.97%** |
| ACMGCN-Moscat | 42.78 ±2.00 | 54.84 ±0.10 | 50.82 ±0.15 | 56.81 ±0.23 | 56.36 ±0.52 | 86.03 ±0.33 | 91.91 ±0.15 | 73.79 ±0.20 | ⇑ **12.28%** |

Table 2: `Moscat` achieves new state-of-the-art results with proper base GNNs. For each dataset, `GNN-Moscat(*)` reports the highest accuracy obtained by selecting its base GNN from GAT, MixHop, GCNII and ACMGCN (see Table 8). Graph MoE baselines are likewise tuned over their respective supported GNN architectures. The top $1^{st}$, $2^{nd}$ and $3^{rd}$ results are highlighted.

| Type | Model | Squirrel | snap-patents | arxiv-year | Flickr | amazon-ratings | Penn94 | genius | ogbn-arxiv |
|---|---|---|---|---|---|---|---|---|---|
| Heterophily GNN | H2GCN | 35.10 ±1.15 | OOM | 49.09 ±0.10 | 51.60 ±0.20 | 46.31 ±0.44 | 81.31 ±0.60 | OOM | 72.80 ±0.24 |
| | GPRGCN | 38.95 ±1.99 | 40.19 ±0.03 | 45.07 ±0.21 | 53.23 ±0.14 | 48.19 ±0.92 | 84.34 ±0.29 | 90.05 ±0.31 | 71.10 ±0.12 |
| | FSGNN | 35.92 ±1.30 | 45.44 ±0.05 | 45.99 ±0.35 | 51.30 ±0.10 | 52.74 ±0.83 | 83.87 ±0.98 | 88.95 ±1.51 | **73.50** ±0.30 |
| | GAT | 39.36 ±1.89 | 44.45 ±0.33 | 52.77 ±0.32 | 55.87 ±0.28 | 49.78 ±0.47 | 81.81 ±0.62 | 88.44 ±1.06 | 72.01 ±0.22 |
| | MixHop | 41.92 ±1.83 | **52.16** ±0.09 | 51.81 ±0.17 | 55.30 ±0.13 | 52.74 ±0.47 | **84.86** ±0.40 | 90.58 ±0.16 | 71.29 ±0.29 |
| | GCNII | 42.48 ±1.86 | 49.18 ±0.23 | 51.75 ±0.36 | **56.31** ±0.27 | 52.45 ±0.57 | 82.34 ±0.51 | 90.41 ±0.35 | 72.74 ±0.17 |
| | ACMGCN | 35.00 ±2.56 | 48.90 ±0.15 | 45.45 ±0.37 | 54.51 ±0.32 | 53.37 ±0.42 | 83.38 ±0.47 | 64.74 ±5.55 | 71.85 ±0.41 |
| Graph Transformer | GraphGPS | 39.81 ±2.28 | OOM | OOM | OOM | 53.27 ±0.66 | OOM | OOM | OOM |
| | SGFormer | **42.65** ±2.41 | 47.74 ±0.15 | 46.63 ±0.20 | 53.48 ±0.13 | 54.14 ±0.62 | 83.07 ±0.49 | 88.47 ±0.43 | 72.76 ±0.33 |
| | Polynormer | 40.17 ±2.11 | OOM | **53.67** ±0.42 | 53.72 ±1.21 | **54.96** ±0.22 | 84.60 ±0.31 | OOM | 73.46 ±0.16 |
| Graph MoE | GMoE | 35.49 ±1.26 | 51.19 ±0.07 | 49.87 ±0.24 | 53.03 ±0.14 | 53.47 ±0.68 | 81.61 ±0.27 | 88.88 ±0.55 | 71.88 ±0.32 |
| | Mowst | 37.75 ±3.73 | 45.38 ±9.56 | 52.56 ±0.22 | 55.48 ±0.32 | 49.13 ±0.64 | 84.56 ±0.31 | 84.80 ±0.54 | 72.52 ±0.07 |
| | DA-MoE | 36.66 ±0.78 | 51.46 ±0.45 | 47.99 ±0.20 | 52.21 ±0.75 | 50.67 ±0.59 | 84.14 ±0.70 | 91.36 ±0.18 | 71.96 ±0.16 |
| Ours | GNN-Moscat∗ | **43.61** ±2.18 | **55.39** ±0.07 | **55.53** ±0.15 | **58.04** ±0.13 | **56.02** ±0.50 | **86.63** ±0.36 | **92.36** ±0.07 | **73.53** ±0.07 |
| | GNN-Moscat | **43.71** ±1.88 | **55.33** ±0.11 | **56.06** ±0.28 | **58.52** ±0.19 | **56.36** ±0.52 | **86.72** ±0.33 | **92.35** ±0.08 | **74.13** ±0.27 |

labeled data in the validation set. We use the complete training set as $\mathcal{V}_{exp}$ and sample 90% of the validation set for $\mathcal{V}_{hold}$, reserving the remaining 10% for gating model validation. While the validation set naturally serves as a holdout set for evaluating model generalization, other baselines can not fully utilize this advantage. Please refer to Appendix F.2 for details about these two setups.

**Baselines and Hyperparameters.** We compare ours against popular homophilous GNNs, heterophilous GNNs, Graph Transformers, Graph MoEs, and also deeper GNNs with various skip-connections. For baseline hyperparameters, we follow their original papers when available; otherwise, we conduct hyperparameter search using Optuna [2]. For `Moscat`, we tune only the gating model's hyperparameters, while experts inherit the same settings as their baseline counterparts. Detailed baseline descriptions and hyperparameters are listed in Appendix F.3 and Appendix F.4.

## 5.2 Performance comparison

**Improvements over GNNs.** To address **Q1**, Table 1 shows that both `Moscat` variants consistently yield substantial improvements across all base GNNs. `Moscat`∗ shows comparable accuracy to `Moscat`, validating that AH's effectiveness does not stem from utilizing more labeled data. Notably, `Moscat` achieves the lowest performance gains with GCNII and the highest with ACM-GCN. This is because GCNII aims to avoid overfitting, limiting AH's impact. Conversely, ACM-GCN is more expressive and prone to overfitting. Table 2 further shows that by selecting proper base GNNs as experts, `Moscat` can outperform state-of-the-art methods by a large margin on all datasets. In contrast, Graph MoEs and Graph Transformers struggle on certain datasets, highlighting the superiority of our paradigm. Appendix G.1 compares with leaderboard results.

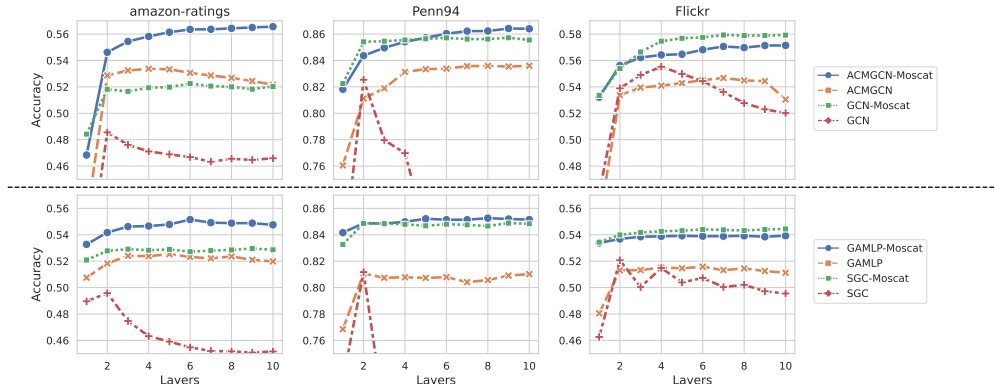

Figure 5: `Moscat` outperforms classic GNNs (e.g., SGC, GCN) and soft-scoping GNNs (e.g., GAMLP, ACMGCN). GAMLP and ACMGCN adaptively learn the scope of SGC and GCN, respectively.

**Compare with techniques for deeper GNNs.** To answer **Q2**, we compare `Moscat` with skip-connection methods for deeper GCN. We examine (1) residual connections (Res: ResNet-like residual connection [23], II: GCNII-like initial residual, Cat: GraphSAGE-like self concatenation [22]) and (2) learnable gating mechanisms (JK: JKNet with concatenation layer aggregation, Attn: GAMLP-like layer aggregation, $G^2$: gradient gating framework [57]). As shown in Table 3, `GCN-Moscat` outperforms other skip-connection techniques on `Penn94`. Additionally, `Moscat` achieves significant improvements over GNNs with skip-connections on `amazon-ratings`. However, skip-connections mix output from different scopes, causing experts to make similar predictions and limiting AH's effectiveness.

Table 3: Accuracy comparison of deeper GCNs (set $L = 6$).

| | Model | amazon-ratings | Penn94 |
|---|---|---|---|
| | GCN | 46.68 ±0.65 | 70.08 ±1.13 |
| | + Moscat | 52.25 ±0.69 | 85.76 ±0.32 |
| Residual Connections | GCN-Res | 47.89 ±0.64 | 82.96 ±0.61 |
| | + Moscat | 51.90 ±0.49 | 85.71 ±0.62 |
| | GCN-II | 51.77 ±0.43 | 80.52 ±0.51 |
| | + Moscat | 54.38 ±0.59 | 85.66 ±0.50 |
| | GCN-Cat | 54.46 ±0.40 | 80.40 ±0.60 |
| | + Moscat | 56.66 ±0.57 | 84.56 ±0.58 |
| Learnable Gating | GCN-JK | 49.45 ±0.37 | 82.80 ±0.48 |
| | + Moscat | 52.23 ±0.88 | 85.69 ±0.27 |
| | GCN-Attn | 49.51 ±0.69 | 82.12 ±0.41 |
| | + Moscat | 52.70 ±0.81 | 85.64 ±0.55 |
| | GCN-$G^2$ | 49.06 ±0.32 | 78.56 ±1.36 |
| | + Moscat | 51.04 ±0.54 | 83.78 ±0.48 |

**Performance variation with depths.** As depth increases, we further investigate how `Moscat` improves two state-of-the-art deeper GNNs with cross-layer gating, ACMGCN [81] and GAMLP [40]. Figure 5 shows that classic GNNs like GCN and SGC often degrade rapidly beyond 2 layers. With gating, GAMLP and ACM-GCN sustain gains up to 4-6 layers. In contrast, `GNN-Moscat` achieves the best results across all depths and continues to improve through 6–10 layers. Notably, when increasing the maximum layers from 1–10, `GNN-Moscat` with message-passing architectures (GCN, ACMGCN) can derive 4–10% accuracy improvements (Figure 5, top), whereas `GNN-Moscat` with decoupled architectures (SGC, GAMLP) yields only 1–2% gains (Figure 5, bottom). This gap likely arises from the higher expressivity of message-passing models, which enables depth-varying experts to produce more diverse predictions. See Appendix G.4 for additional experiments.

## 5.3 Ablation study

In this subsection, we evaluate each component in `Moscat` to answer **Q3**. As shown in Table 4, using Mean-Ensemble to average GNN logits across different depths cannot guarantee accuracy improvements, primarily due to the poor performance of deeper GNNs. By enabling node-level adaptation, `Moscat` consistently achieves significant improvements over Mean-Ensemble. We evaluate the effectiveness of critical components in `Moscat` by removing them one at a time: (1) When we remove the holdout set and train both the gating model and experts on the full training set, we observe a drastic accuracy drop compared to `Moscat*`

Table 4: Ablation study. *Mean-Ensemble* combines all scope experts with uniform weights. For `Moscat`, *w/o Holdout-Set* trains the gating model on the same expert training set, *w/o Multi-Scope* ensemble experts with the same best scope, *w/o Hetero-Filter* removes heterophily-biased sample filtering, and *w/o Scope-Augment* removes scope-aware logit augmentation.

| | Squirrel | amazon-ratings | Penn94 | arxiv-year |
|---|---|---|---|---|
| SGC | 40.04 ±1.77 | 49.58 ±0.55 | 81.17 ±0.40 | 45.88 ±0.32 |
| w/ Mean-Ensemble | 39.65 ±1.76 | 50.56 ±0.53 | 80.20 ±0.77 | 46.68 ±0.25 |
| SGC-Moscat∗ | 42.73 ±2.06 | 51.82 ±0.29 | 84.75 ±0.41 | 52.09 ±0.27 |
| SGC-Moscat | 43.35 ±1.97 | 52.72 ±0.53 | 84.80 ±0.49 | 51.86 ±0.30 |
| w/o Holdout-Set | 39.86 ±2.31 | 49.56 ±0.31 | 77.72 ±0.95 | 49.80 ±0.23 |
| w/o Multi-Scope | 40.98 ±1.72 | 51.09 ±0.56 | 82.20 ±0.45 | 46.63 ±0.28 |
| w/o Hetero-Filter | 42.48 ±2.37 | 50.87 ±0.88 | 84.64 ±0.52 | - |
| w/o Scope-Augment | 42.29 ±2.06 | 52.34 ±0.64 | 82.79 ±0.49 | 47.10 ±0.28 |

(which also restricts model training within the training set). This highlights the necessity of the holdout set. (2) Multi-Scope experts prove critical for incorporating knowledge from different scopes. (3) Heterophily-biased sample filtering is designed as an optional technique, where "-" denotes it not being used in our hyperparameter settings. We find it critical for sampling high-quality data for gating model training. (4) Scope-aware logit augmentation shows particular effectiveness when combined with SGC, likely due to SGC's limited model expressivity.

### 5.4 Case Study: how `Moscat` become effective?

We examine the >40% performance gain of `ACMGCN-Moscat` on `genius` (Table 1). Dropping the MLP expert and using only six weak experts (1–6 layers ACMGCN) yields the same improvement, so the strong expert isn't driving this effect. Figure 6 (layers 1–2 omitted) shows that: (1) Even mistaken ACMGCN predictions carry structurally informative logits. (2) When there are sufficient training samples, `Moscat` can learn

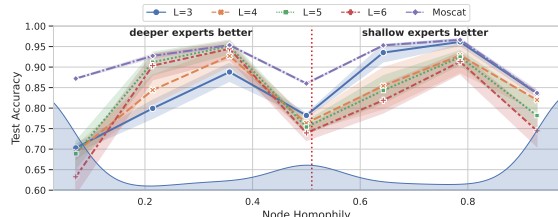

Figure 6: `Moscat` over ACMGCN on `genius`.

proper expert mixtures for correct predictions. (3) With limited samples, `Moscat` can still learn to select the best-performing expert.

## 6 Conclusion

In this paper, we investigate the challenges of applying deeper GNNs to heterophilous graphs. Our analysis reveals that GNNs exhibit shifting generalization preferences across nodes with different homophily levels as their depth increases. To address this, we propose `Moscat`, which follows a novel decoupled expert-gating paradigm. Experiments show that `Moscat` can improve GNN generalization, better exploit deeper GNNs, and adapt to diverse architectures.

## Acknowledgments and Disclosure of Funding

This work is supported by DEVCOM ARL Army Research Office (ARO) under grants W911NF2220159 and W911NF2320186, and by National Science Foundation (NSF) under grant OAC-2209563. Distribution Statement A: Approved for public release. Distribution is unlimited.

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

# A Related Work

## A.1 Graph neural networks meet heterophily

GNNs were initially designed under the homophily assumption and have recently gained significant interest due to their superior performance and small parameterization. Various aspects of GNNs have been widely studied, including scalability [22, 76, 5, 20], expressivity [69, 38, 51], and generalization [13, 74, 45].

To extend GNNs to heterophilous graphs, existing works primarily focus on improving higher-order neighborhood utilization [71]. MixHop [1] extracts features from multi-hop neighborhoods in each layer. GCNII [8] prevents over-smoothing in deeper GCNs by proposing initial residual connections and identity mapping. To adapt to graphs with different label patterns, GPR-GNN [11] learns signed scalar weights for the propagated features with different propagation steps. Other works focus on mapping topology [36, 33, 40], using global attentions [15], or exploiting edge directionality [56] to improve learning on heterophilous graphs.

## A.2 GNNs with personalized scoping

Personalized scoping aims to set a tailored receptive field to each node to restrict the length of feature propagation, which is able to extract essential long-range dependencies while reducing computational overhead. Although this idea has been around for a long time and has a fundamental impact on various GNN domains, few works summarize these advancements from the personalized scoping perspective. Here, we classify these works into two categories based on whether or not each scope is learned.

**Heuristics methods.** The heuristics for personalized scoping originate from works that generalize personalized PageRank (PPR) to GNNs. PPNP [19] first introduces PPR as the final propagation matrix for decoupled GNNs. GBP [7] combines reverse push and random walks to approximate PPR propagation. NDLS [79] examines the smoothing effects in graph diffusion, noting that the level of smoothness should be node-specific. NDM [24] advances this by developing a unified diffusion kernel that extends PPR with the heat kernel and enables custom propagation steps following NDLS. To generalize personalized scoping to non-decoupled GNNs, ShaDow [77] proposes a design principle that decouples the scope from the model depth. For each node, a shallow scope is constructed using its neighboring nodes with the top-k PPR.

**Learnable methods.** However, these heuristics assume homophily and heavily rely on topological information, often falling short on heterophilous graphs. Recent research has explored parameterized techniques to address this issue. One line of work integrates personalized scoping into GNN architectures. We refer to these methods as soft personalized scoping since they typically learn node-dependent weights to control the scope. GeniePath [37] proposes a gated unit as the scope controller. GAMLP [81] uses the attention mechanism to enable personalized scoping on a decoupled GNN [16]. NW-GNN [63] further extends this to non-decoupled GNNs by proposing a node-wise architectural search. ACM-GCN [40] employs an alternative strategy by introducing additional identity channels beyond aggregation.

## A.3 Graph Mixture of Experts

Another line of work considers different depths of GNNs as different experts and develops a gating module to activate a small subset of experts for each input node. Policy-GCN [29] uses reinforcement techniques and takes the average accuracy of different depth models as rewards. GraphMoE [61] and DA-MoE [75] proposes a top-$K$ sparse gating technique for mixing multiple GNN experts. Additionally, Mowst [78] proposes separating rich self-features from informative neighborhoods by using a mixture of weak MLP and strong GNN experts. However, these methods require retraining the GNN models, which introduces significant overhead and may suffer from overfitting, potentially downgrading each model's performance.

# B Proof

The following provides the proof for Theorem 3.3.

**Lemma B.1.** *Under Assumption 3.1, Assumption 3.2, and assume the aggregated features $g^L(\mathbf{X}, \mathcal{G})$ share the same variance $\sigma^2 \mathbf{I}$. For any subgroup $i, j \in M$ and any nodes $u \in \mathcal{V}_i, v \in \mathcal{V}_j$ with aggregated features $\mathbf{f}_u = g^L(\mathbf{X}, \mathcal{G})_u$ and $\mathbf{f} = g^L(\mathbf{X}, \mathcal{G})_v$, we have:*

$$\Pr(y_u = c_1 | \mathbf{f}_u) - \Pr(y_v = c_1 | \mathbf{f}_v)$$
$$\leq \frac{2(\rho + k\sigma)}{\sigma^2} \left( \|\mathbf{f}_u - \mathbf{f}_v\| + \rho(p_i - p_j) \mathbb{E}_{\Pr(m)} [p_m - q_m]^{L-1} \right), \tag{5}$$

*where $\rho = \|\boldsymbol{\mu}_1 - \boldsymbol{\mu}_2\|$ and $k$ is a small constant.*

*Proof.* To begin with, we recall that Assumption 3.1 assumes every node feature follows the normal distribution. The aggregated features $\mathbf{F} = g^L(\mathbf{X}, \mathcal{G}) = \left(\mathbf{D}^{-1}\mathbf{A}\right)^L \mathbf{X}$ for different subgroups of different classes has the following distribution:

$$\mathbf{f}_w \sim N\left(\boldsymbol{\mu}_{c,m}^{(L)}, \sigma^2 \mathbf{I}\right), \text{ for } w \in \mathcal{V}_m^c, \ c \in \{1, 2\}, m \in \{1, \cdots, M\}, \tag{6}$$

where $\mathcal{V}_m^c$ denotes the node subset with class label $c$ and belongs to subgroup $m$. $\boldsymbol{\mu}_{c,m}^{(L)}$ denotes the mean value of $L$-hop aggregated features of $\mathcal{V}_m^c$.

Next, we break down $\Pr(y_u = c_1 | \mathbf{f}_u)$, which is the conditional probability of node $u \in \mathcal{V}_i$ classified as class $c_1$ given feature $\mathbf{f}_u$, with Bayes theorem

$$\Pr(y_u = c_1 | \mathbf{f}_u)$$
$$= \frac{\Pr(\mathbf{f}_u | y_u = c_1) \Pr(y_u = c_1)}{\Pr_1(\mathbf{f}_u | y_u = c_1) \Pr(y_u = c_1) + \Pr_1(\mathbf{f}_u | y_u = c_2) \Pr(y_u = c_2)}$$
$$\overset{(a)}{=} \frac{\exp\left(\frac{(\mathbf{f}_u - \boldsymbol{\mu}_{1,i}^{(L)})^2}{-2\sigma^2}\right)}{\exp\left(\frac{(\mathbf{f}_u - \boldsymbol{\mu}_{1,i}^{(L)})^2}{-2\sigma^2}\right) + \exp\left(\frac{(\mathbf{f}_u - \boldsymbol{\mu}_{2,i}^{(L)})^2}{-2\sigma^2}\right)}, \tag{7}$$

where (a) utilize Assumption 3.1 that different classes have the same number of samples and the PDF of normal distribution. Hence, we have

$$\Pr(y_u = c_1 | \mathbf{f}_u) - \Pr(y_u = c_1 | \mathbf{f}_v) =$$
$$\frac{\exp\left(\frac{(\mathbf{f}_u - \boldsymbol{\mu}_{1,i}^{(L)})^2}{-2\sigma^2}\right) \exp\left(\frac{(\mathbf{f}_v - \boldsymbol{\mu}_{2,j}^{(L)})^2}{-2\sigma^2}\right) - \exp\left(\frac{(\mathbf{f}_v - \boldsymbol{\mu}_{1,j}^{(L)})^2}{-2\sigma^2}\right) \exp\left(\frac{(\mathbf{f}_u - \boldsymbol{\mu}_{2,i}^{(L)})^2}{-2\sigma^2}\right)}{\left[\exp\left(\frac{(\mathbf{f}_u - \boldsymbol{\mu}_{1,i}^{(L)})^2}{-2\sigma^2}\right) + \exp\left(\frac{(\mathbf{f}_u - \boldsymbol{\mu}_{2,i}^{(L)})^2}{-2\sigma^2}\right)\right]\left[\exp\left(\frac{(\mathbf{f}_v - \boldsymbol{\mu}_{1,j}^{(L)})^2}{-2\sigma^2}\right) + \exp\left(\frac{(\mathbf{f}_v - \boldsymbol{\mu}_{2,j}^{(L)})^2}{-2\sigma^2}\right)\right]}, \tag{8}$$

Note that the denominator can be bounded in $[0, 4]$ since each component $\exp\left(\frac{(\mathbf{f}_w - \boldsymbol{\mu}_{c,m}^{(L)})^2}{-2\sigma^2}\right) \in [0, 1]$.

We can then denote the denominator as $\exp(A)$, where A is a constant. Hence, we have

$$\Pr(y_u = c_1 | \mathbf{f}_u) - \Pr(y_v = c_1 | \mathbf{f}_v)$$
$$= \exp\left(\frac{(\mathbf{f}_u - \boldsymbol{\mu}_{1,i}^{(L)})^2 + (\mathbf{f}_v - \boldsymbol{\mu}_{2,j}^{(L)})^2}{-2\sigma^2} - A\right)$$
$$- \exp\left(\frac{(\mathbf{f}_v - \boldsymbol{\mu}_{1,j}^{(L)})^2 + (\mathbf{f}_u - \boldsymbol{\mu}_{2,i}^{(L)})^2}{-2\sigma^2} - A\right)$$
$$\overset{(a)}{\leq} \frac{1}{2\sigma^2}\left[(\mathbf{f}_v - \boldsymbol{\mu}_{1,j}^{(L)})^2 - (\mathbf{f}_u - \boldsymbol{\mu}_{1,i}^{(L)})^2 + (\mathbf{f}_u - \boldsymbol{\mu}_{2,i}^{(L)})^2 - (\mathbf{f}_v - \boldsymbol{\mu}_{2,j}^{(L)})^2\right]$$
$$= \frac{1}{2\sigma^2}\left[\left((\mathbf{f}_v - \boldsymbol{\mu}_{1,j}^{(L)}) + (\mathbf{f}_u - \boldsymbol{\mu}_{1,i}^{(L)})\right)\left((\mathbf{f}_v - \mathbf{f}_u) + (\boldsymbol{\mu}_{1,i}^{(L)} - \boldsymbol{\mu}_{1,j}^{(L)})\right)\right.$$
$$\left. + \left((\mathbf{f}_v - \boldsymbol{\mu}_{2,j}^{(L)}) + (\mathbf{f}_u - \boldsymbol{\mu}_{2,i}^{(L)})\right)\left((\mathbf{f}_u - \mathbf{f}_v) + (\boldsymbol{\mu}_{2,j}^{(L)} - \boldsymbol{\mu}_{2,i}^{(L)})\right)\right] \tag{9}$$

(a) is derived from the Lagrange mean value theorem. Let $(\mathbf{f}_u - \boldsymbol{\mu}_{1,i}^{(L)})^2 + (\mathbf{f}_v - \boldsymbol{\mu}_{2,j}^{(L)})^2 = C$ and $(\mathbf{f}_v - \boldsymbol{\mu}_{1,j}^{(L)})^2 + (\mathbf{f}_u - \boldsymbol{\mu}_{2,i}^{(L)})^2 = D$. Given that $\Pr(y_u = c_1|\mathbf{f}_u)$ and $\Pr(y_v = c_1|\mathbf{f}_v)$ are probabilities that smaller than 1, we can derive

$$\frac{C}{-2\sigma^2} - A < 0 \text{ and } \frac{D}{-2\sigma^2} - A < 0. \tag{10}$$

From the Lagrange mean value theorem, we have

$$\frac{\exp(x) - \exp(y)}{x - y} = \exp(\xi), \ \xi \in (x, y). \tag{11}$$

Let $x = \frac{C}{-2\sigma^2} - A$ and $y = \frac{D}{-2\sigma^2} - A$. Given Equation 12, we have $\exp(\xi) \le 1$. Hence,

$$\exp\left(\frac{C}{-2\sigma^2} - A\right) - \exp\left(\frac{D}{-2\sigma^2} - A\right) \le \frac{D - C}{2\sigma^2}. \tag{12}$$

The proof for (a) is complete.

Equation 9 shows that $\Pr(y_u = c_1|\mathbf{f}_u) - \Pr(y_v = c_1|\mathbf{f}_v)$ can be bounded by three terms:

1. The maximum distance between any node feature to the mean value of any subgroup

$$\epsilon_{ij} = \max_{w \in \mathcal{V}_i \cup \mathcal{V}_j, c \in \{1,2\}, m \in \{i,j\}} \|\mathbf{f}_w - \boldsymbol{\mu}_{c,m}^{(L)}\|. \tag{13}$$

2. The feature distance $\|\mathbf{f}_u - \mathbf{f}_v\|$ between node $u$ and $v$.

3. The mean value difference between subgroup $i$ and $j$: $\boldsymbol{\mu}_{1,i}^{(L)} - \boldsymbol{\mu}_{1,j}^{(L)}$ and $\boldsymbol{\mu}_{2,i}^{(L)} - \boldsymbol{\mu}_{2,j}^{(L)}$.

We first bound term (1) $\epsilon_{ij}$. Recall that $\mathbf{f}_w$ follows the normal distribution with variance $\sigma^2$. For the mean value of the aggregated feature of node set $\mathcal{V}_i^1$ and $\mathcal{V}_i^2$, we have

$$\boldsymbol{\mu}_{1,i}^{(L)} = p_i \mathbb{E}_{\Pr(m)}\left[\boldsymbol{\mu}_{1,m}^{(L-1)}\right] + q_i \mathbb{E}_{\Pr(m)}\left[\boldsymbol{\mu}_{2,m}^{(L-1)}\right], \tag{14}$$

and

$$\boldsymbol{\mu}_{2,i}^{(L)} = q_i \mathbb{E}_{\Pr(m)}\left[\boldsymbol{\mu}_{1,m}^{(L-1)}\right] + p_i \mathbb{E}_{\Pr(m)}\left[\boldsymbol{\mu}_{2,m}^{(L-1)}\right], \tag{15}$$

which reveals that for every $L$ we have

$$\min(\boldsymbol{\mu}_1, \boldsymbol{\mu}_2) \le \boldsymbol{\mu}_{1,i}^{(L)}, \boldsymbol{\mu}_{2,i}^{(L)} \le \max(\boldsymbol{\mu}_1, \boldsymbol{\mu}_2) \tag{16}$$

since every $p_i, q_i, \Pr(m)$ lie in $[0, 1]$. Therefore, $\epsilon_{ij}$ can be bounded by

$$\epsilon_{ij} \le \|\boldsymbol{\mu}_1 - \boldsymbol{\mu}_2\| + k\sigma, \tag{17}$$

where k is a small constant. When $k > 3$, this equation holds with a probability close to 1.

To bound the term (3), we derive the following equation based on Equation 14 and Equation 15

$$
\begin{aligned}
&\boldsymbol{\mu}_{1,i}^{(L)} - \boldsymbol{\mu}_{2,i}^{(L)} \\
&= (p_i - q_i) \cdot \mathbb{E}_{\Pr(m)}\left[\boldsymbol{\mu}_{1,m}^{(L-1)}\right] - (p_i - q_i) \cdot \mathbb{E}_{\Pr(m)}\left[\boldsymbol{\mu}_{2,m}^{(L-1)}\right] \\
&= (p_i - q_i) \cdot \mathbb{E}_{\Pr(m)}\left[\boldsymbol{\mu}_{1,m}^{(L-1)} - \boldsymbol{\mu}_{2,m}^{(L-1)}\right] \\
&= (p_i - q_i) \cdot \mathbb{E}_{\Pr(m)}\left[(p_m - q_m) \cdot \mathbb{E}_{\Pr(o)}\left[\boldsymbol{\mu}_{1,o}^{(L-2)} - \boldsymbol{\mu}_{2,o}^{(L-2)}\right]\right] \\
&= (p_i - q_i) \cdot \mathbb{E}_{\Pr(m)}\left[p_m - q_m\right]^1 \cdot \mathbb{E}_{\Pr(m)}\left[\boldsymbol{\mu}_{1,m}^{(L-2)} - \boldsymbol{\mu}_{2,m}^{(L-2)}\right] \\
&= (p_i - q_i) \cdot \mathbb{E}_{\Pr(m)}\left[p_m - q_m\right]^{L-1} \cdot (\boldsymbol{\mu}_1 - \boldsymbol{\mu}_2).
\end{aligned}
\tag{18}
$$

We also have

$$\boldsymbol{\mu}_{1,i}^{(L)} - \boldsymbol{\mu}_{1,j}^{(L)}$$

$$= (p_i - p_j) \cdot \mathbb{E}_{\mathrm{Pr}(m)} \left[ \boldsymbol{\mu}_{1,m}^{(L-1)} \right] + (q_i - q_j) \cdot \mathbb{E}_{\mathrm{Pr}(m)} \left[ \boldsymbol{\mu}_{2,m}^{(L-1)} \right]$$

$$\underset{(a)}{=} (p_i - p_j) \cdot \mathbb{E}_{\mathrm{Pr}(m)} \left[ \boldsymbol{\mu}_{1,m}^{(L-1)} - \boldsymbol{\mu}_{2,m}^{(L-1)} \right] \tag{19}$$

$$= (p_i - p_j) \cdot \mathbb{E}_{\mathrm{Pr}(m)} \left[ (p_m - q_m) \cdot \mathbb{E}_{\mathrm{Pr}(o)} \left[ p_o - q_o \right]^{L-2} (\boldsymbol{\mu}_1 - \boldsymbol{\mu}_2) \right]$$

$$= (p_i - p_j) \cdot \mathbb{E}_{\mathrm{Pr}(m)} \left[ p_m - q_m \right]^{L-1} (\boldsymbol{\mu}_1 - \boldsymbol{\mu}_2),$$

where (a) utilizes the definition of $p_i + q_i = 1$. Follow the same proof, we can derive $\boldsymbol{\mu}_{2,i}^{(L)} - \boldsymbol{\mu}_{2,j}^{(L)} = -(\boldsymbol{\mu}_{1,i}^{(L)} - \boldsymbol{\mu}_{1,j}^{(L)})$.

Next, we substitute terms (1), (2), and (3) to Equation 9 to continue the proof. Let $\rho = \|\boldsymbol{\mu}_1 - \boldsymbol{\mu}_2\|$, we have

$$\mathrm{Pr}(y_u = c_1 | \mathbf{f}_u) - \mathrm{Pr}(y_v = c_1 | \mathbf{f}_v)$$

$$\leq \frac{1}{2\sigma^2} \left( (\mathbf{f}_v - \boldsymbol{\mu}_{1,j}^{(L)}) + (\mathbf{f}_u - \boldsymbol{\mu}_{1,i}^{(L)}) + (\mathbf{f}_v - \boldsymbol{\mu}_{2,j}^{(L)}) + (\mathbf{f}_u - \boldsymbol{\mu}_{2,i}^{(L)}) \right)$$

$$\cdot \left( \|\mathbf{f}_u - \mathbf{f}_v\| + (p_i - p_j) \cdot \mathbb{E}_{\mathrm{Pr}(m)} \left[ p_m - q_m \right]^{L-1} (\boldsymbol{\mu}_1 - \boldsymbol{\mu}_2) \right) \tag{20}$$

$$\leq \frac{2(\rho + k\sigma)}{\sigma^2} \left( \|\mathbf{f}_u - \mathbf{f}_v\| + \rho(p_i - p_j) \cdot \mathbb{E}_{\mathrm{Pr}(m)} \left[ p_m - q_m \right]^{L-1} \right).$$

The proof for Lemma B.1 is complete. $\qquad \square$

**Proof for Theorem 3.3.** Theorem 1 in [45] provides a PAC-Bayes subgroup generalization bound for GNNs with one-hop aggregation. In our work, we extend their theorem to arbitrary hop to investigate how different scope sizes affect the generalization of different subgroups. Besides that, both theorem follows the same setting. To avoid replicates, we only provide the proof for the different parts and direct readers to the specific sections of [45] for the rest details and proofs.

Appendix F.4 in [45] provides complete proof for the proposed PAC-Bayes bound. There exists $0 < \alpha < \frac{1}{4}$, we have

$$\mathcal{L}_m^0(\theta) - \widehat{\mathcal{L}}_S^\gamma(\theta)$$

$$\leq \left( D_{m,0}^{\gamma/2}(P; \lambda) - \ln 3 \right) + \frac{d \sum_{l=1}^{L'} \|\widetilde{W}_l\|_F^2}{(\gamma/8)^{2/L'} N_S^\alpha} (\epsilon_m)^{2/L'} \tag{21}$$

$$+ \frac{1}{N_S^{2\alpha}} \left( \ln \frac{3}{\delta} + 2 \right) + \frac{1}{4 N_S^{1-2\alpha}},$$

where $D_{m,0}^{\gamma/2}(P; \lambda)$ is bounded by [45]'s Lemma 4.

Lemma 4 in [45] further depends on the bound of [45]'s Lemma 2 in Appendix E, which calls out the main difference compared with our theorem. By substituting [45]'s Lemma 2 with our Lemma B.1, we complete the proof for our Theorem 3.3

$$\mathcal{L}_m^0(\theta) - \widehat{\mathcal{L}}_S^\gamma(\theta)$$

$$\leq \frac{2K(\rho + k\sigma)}{\sigma^2} \left( \epsilon_m + \rho(p_S - p_m) \mathbb{E}_{\mathrm{Pr}(o)} \left[ p_o - q_o \right]^{L-1} \right) \tag{22}$$

$$+ \frac{d \sum_{l=1}^{L'} \|\widetilde{W}_l\|_F^2}{(\gamma/8)^{2/L'} N_S^\alpha} (\epsilon_m)^{2/L'} + \frac{1}{N_S^{2\alpha}} \left( \ln \frac{3}{\delta} + 2 \right) + \frac{1}{4 N_S^{1-2\alpha}}.$$

For the first term, $K = 2$ is the number of classes, $k$ is a small constant, and $\sigma < 1$. We denote $\mathbb{E}_{\mathrm{Pr}(o)} \left[ p_o - q_o \right]^{L-1}$ as $\Gamma_{L-1}$. For the rest of the terms, $b$ and $\gamma$ are constants. We denote $\|W\|^2 := \sum_{l=1}^{L'} \|\widetilde{W}_l\|_F^2$. Since $0 < \alpha < \frac{1}{4}$, we have $\frac{1}{4 N_S^{1-2\alpha}} < 1$. We simplify Equation 22 as follows:

$$\mathcal{L}_m^0(\theta) - \widehat{\mathcal{L}}_S^\gamma(\theta)$$

$$\leq \mathcal{O}\Big(\frac{\rho}{\sigma^2}\Big(\epsilon_m \, + \, \rho\,(p_S - p_m)\,\Gamma_{L-1}\Big)\Big) + \mathcal{O}\Big(\frac{\|W\|^2\,(\epsilon_m)^{2/L'}}{N_S^\alpha}\Big) + \mathcal{O}\Big(\frac{\ln(1/\delta)}{N_S^{2\alpha}}\Big). \qquad (23)$$

## C  Supplementary Preliminaries

### C.1  Scope and Depth

The scope of model $\mathcal{M}$ for a node $v$ is a latent subgraph $\mathcal{G}_{[v]}^{\mathcal{M}}$ that contains all the nodes $\mathcal{V}_{[v]} \subseteq \mathcal{V}$ and edges $\mathcal{E}_{[v]} \subseteq \mathcal{E}$ used by $\mathcal{M}$ when predicting $v$.

**Definition C.1** (Size of scope). Assume $\mathcal{G}_{[v]}^{\mathcal{M}}$ is connected. The size of scope $\left|\mathcal{G}_{[v]}^{\mathcal{M}}\right| = \max_{u \in \mathcal{V}_{[v]}} d\,(u, v)$, where $d\,(u, v)$ denotes the shortest path distance from $u$ to $v$.

We focus on GNN architectures where the model depth equals the size of the scope for every node in this paper.

### C.2  Homophily Metrics

The homophily/heterophily metrics are widely used as graph properties to measure the probability of nodes with the same class connected to each other. This paper uses a node-wise homophily metric called *node homophily* [52]. Node homophily defines the fraction of neighbors that have the same class for each node:

$$h_{\text{node}}[v] = |\{u \in \mathcal{N}_v : y_u = y_v\}|/|\mathcal{N}_v| \qquad (24)$$

In Table 2, we also report the average node homophily for each dataset:

$$H_{\text{node}}^{\mathcal{G}} = \frac{1}{|\mathcal{V}|} \sum_v h_{\text{node}}[v]. \qquad (25)$$

There are also some other commonly used homophily metrics in the literature. *Edge homophily* [1, 83] measures the fraction of edges that connect nodes of the same class. *Class homophily* [36, 40] further addresses the sensitivity issue of edge homophily on graphs with imbalanced classes. For instance, `genius` dataset [36] has a majority class for roughly 80% of nodes, which can mislead edge homophily into classifying it as a homophilous graph ($H_{\text{edge}}^{\mathcal{G}} = 0.618$). In contrast, class homophily accurately identifies it as a heterophilous graph ($H_{\text{class}}^{\mathcal{G}} = 0.080$). However, class homophily does not satisfy some desired properties, such as asymptotic constant baseline and empty class tolerance. [54] proposed a variant called *adjusted homophily* to address this issue.

### C.3  GNN Basics.

As a pioneer work, graph convolutional network (GCN) [28] provides a layer-wise architecture that stacks feature propagation with linear transformation to approximate spectral graph convolutions. The $(l+1)-$th layer of GCN is defined as:

$$\boldsymbol{h}_v^{(\ell+1)} = \texttt{ReLU}\left(\sum_{u \in \mathcal{N}_v \cup \{v\}} \widetilde{\mathbf{A}}_{v,u} \boldsymbol{h}_u^{(\ell)} \mathbf{W}^{(\ell)}\right), \qquad (26)$$

where $\mathbf{h}_u^{(0)} = \boldsymbol{x}_u$. Subsequent GNNs typically modify GCN in terms of aggregator and backbone. Some works develop more expressive aggregators [60, 14, 34]. For example, GAT [60] substituting the degree-normalized coefficients in GCN's aggregator with learnable attention scores. Another line of works alters the backbone by decoupling aggregation from transformation [65, 19, 16, 81, 46] or adding skip connections between layers [68, 31, 8, 80]. For instance, SGC [65] eliminates the $\texttt{ReLU}\,(\cdot)$ function in the GCN layer to allow precomputing feature propagation. JKNet [68] directly maps the concatenated outputs of each layer $[\mathbf{h}_v^{(0)}; \cdots ; \mathbf{h}_v^{(L)}]$ to the prediction.

## C.4 Personalized Scoping.

Real-world graphs often exhibit a mix of homophilous and heterophilous patterns [36, 33, 45]. An ideal way to incorporate sufficient homophilous information is to set a *personalized scope*, allowing different nodes to have distinct scope sizes.

**Definition C.2** (Model with personalized scoping). Assume the input graph $\mathcal{G}$ is connected with radius exceed $\max_{v \in \mathcal{V}} \left| \mathcal{G}_{[v]}^{\mathcal{M}} \right|$. The model $\mathcal{M}$ has personalized scoping if $\exists u, v \in \mathcal{V}$, where $\left| \mathcal{G}_{[u]}^{\mathcal{M}} \right| \neq \left| \mathcal{G}_{[v]}^{\mathcal{M}} \right|$.

The above definitions assume the graph's radius is larger than each node's scope size, which is true for most nodes in large graphs. According to the definition, an $L$-layer GCN does not support personalized scoping because it forces the scope size to be uniformly $L$ for every node. To break this limit, recent works (e.g., ACMGCN, GAMLP, NW-GNN, GNN-G$^2$) propose additional gating modules that learn the weight $\alpha_v^{(\ell)}$ for each hop on a node-dependent basis:

$$\boldsymbol{h}_v^{\text{out}} = \sum_{\ell=0}^{L} \alpha_v^{(\ell)} \boldsymbol{h}_v^{(\ell)}, \text{ where } \sum_{\ell=0}^{L} \alpha_v^{(\ell)} = 1 \tag{27}$$

By setting the weights of larger scopes to 0 for some nodes, these methods allow different nodes to have different scope sizes. In this paper, we adopt a broader definition: we consider a model to support personalized scoping if it can generate node-adaptive weights for each scope embedding. Notably, our proposed method `Moscat` also enables personalized scoping for any base GNN architecture through an attention-based gating mechanism.

$$\mathcal{L}_m^0(\theta) - \widehat{\mathcal{L}}_S^\gamma(\theta) \leq \mathcal{O}\left( \frac{\rho}{\sigma^2} \left( \epsilon_m + \rho \left( p_S - p_m \right) \Gamma_{L-1} \right) \right) + \mathcal{O}\left( \frac{\|W\|^2 \left( \epsilon_m \right)^{2/L'}}{N_S^\alpha} \right) + \mathcal{O}\left( \frac{\ln(1/\delta)}{N_S^{2\alpha}} \right) \tag{28}$$

# D   Supplementary Analysis for Deeper GNNs' Failure

**The depth dilemma.** Increasing the GNN depth favorably increases the number of non-linear transformations and expands the scope of each node to exponentially incorporate more neighboring information. Existing studies also show that deeper GNNs are provably more expressive by using distance-based metrics [50, 25] and WL-based metrics [48, 10]. In practice, however, performance degradation is widely observed when going deep. Many novel architectures have been proposed to alleviate this issue, yet they achieve only marginal gains over shallow variants, which is undesirable given deeper GNNs' superior expressive power and substantially higher computational costs. How to effectively leverage the depth remains an open question.

**Limitation of existing solutions.** While useful, current techniques for deeper GNNs face inherent trading-offs between these three types of errors. Residual connections (e.g., ResGCN) address the vanishing gradient issue but result in a larger generalization gap [13]. Learnable cross-layer gating modules (e.g., G$^2$-GNN) achieve personalized scoping to alleviate over-smoothing, but they lead to an increase in training difficulties and the risk of overfitting. Meanwhile, GCNII attempts to prevent overfitting by mixing initial embeddings and adding identity mappings to weight matrices, but this comes at the cost of reduced expressivity.

# E   Compare Expert-Gate Joint Training MoEs with Independent Train-then-Merge MoEs

Expert-gate joint training is widely used in current MoE methods for LLMs. Instead, `Moscat` employs a separate "train-then-merge" strategy which is optimized for GNN experts. Our work is motivated by the observation that directly transferring MoE designs from LLMs to GNNs faces inherent challenges. In LLMs, the gating module and experts are co-trained using Top-K sparse gating to reduce training and inference time while maintaining performance. However, for GNNs, no clear scaling law exists, and sparse gating plus more parameters do not necessarily translate into accuracy improvements. In our experiments with `Moscat`, we found that Top-K sparse gating (especially with small K) often results in an accuracy drop compared to dense gating. Recent work [9] also reveals that sparse

Table 5: Hyperparameter searching space for GNNs.

| Hyperparameters | Range |
|---|---|
| learning rate | { 0.05, 0.01, 0.005, 0.001, 5e-4, 1e-4, 5e-5 } |
| normalization | { layer [3], batch [26], - } |
| hidden dimension | { 32, 64, 128, 256, 512 } |
| dropout | { 0, 0.1, 0.3, 0.5, 0.7 } |
| number of convolution layers | { 1, 2, 3, 4, 5, 6 } |

Table 6: Hyperparameter Searching space for `Moscat`.

| | Hyperparameters | Range | Notes |
|---|---|---|---|
| Moscat specific hyperparameters | maximum scope size $L_{\max}$ | { 5, 6 } | increasing $L_{\max} > 6$ can slightly improve accuracy |
| | validation ratio $\eta$ | { 0, 0.1 } | dataset specific parameter |
| | The $\mathcal{V}_{\text{tr-het}}$ masked ratio $\gamma$ | { 0, 0.3, 0.5, 0.9, 1, - } | "-" indicates $\mathcal{V}_{\exp}$ is not used for `Moscat` training |
| | mask wrong | { True, False } | - |
| hyperparameters for MLP in Moscat | learning rate | { 0.005, 0.001, 5e-4, 1e-4, 5e-5 } | - |
| | normalization | { layer, batch, - } | dataset specific parameter |
| | hidden dimension | { 64, 128, 256 } | fix to 256 for larger datasets (if not OOM) |
| | dropout | { 0 } | fix dropout to 0 works the best |
| | number of layers | { 3 } | 3-layer MLP is sufficient for all settings |

gating can lead to performance losses on heterophilous datasets. Our independent-train-then-merge approach for GNN MoEs offers distinct advantages over expert-gate joint training:

- **Diverse Generalization Capabilities:** GNN experts, which vary in scope and architecture, exhibit substantial disparities in generalization capability, which has been largely overlooked. Graph data spans a wide range of domains from different perspectives (e.g., homophily, centrality). By training experts independently, we allow each to specialize and become robust domain generalizers.

- **Avoiding Harmful Regularization:** Standard joint training requires regularization to balance node distribution among experts and prevent collapse. However, graph domains are often unevenly distributed, typically following a power-law. Imposing such regularization can diminish the expressive power of the GNN experts. Our experimental results (Table 2) demonstrate that existing GNN MoEs underperform even well-tuned single GNNs, whereas `Moscat` successfully learns meaningful gating weights (see Figures 18 and 19) without compromising the underlying GNN training.

In summary, we highlight the limitations of directly applying LLM MoE training strategies to GNNs and demonstrate how `Moscat` provides a more promising alternative that better leverages the unique properties of graph data. We believe `Moscat` will provide valuable insights to the community and open new directions for advancing GNN capabilities.

Table 7: Values of the hyperparameter $\alpha$ used in the `Moscat∗` setting reported in Table 1.

| | Squirrel | snap-patents | arxiv-year | Flickr | amazon-ratings | Penn94 | genius | ogbn-arxiv |
|---|---|---|---|---|---|---|---|---|
| SGC-Moscat∗ | 0.6 | 0.65 | 0.65 | 0.85 | 0.9 | 0.9 | 0.9 | 0.85 |
| GCN-Moscat∗ | 0.55 | 0.65 | 0.65 | 0.85 | 0.9 | 0.85 | 0.9 | 0.9 |
| GAT-Moscat∗ | 0.55 | 0.65 | 0.65 | 0.85 | 0.9 | 0.85 | 0.9 | 0.85 |
| GCNII-Moscat∗ | 0.6 | 0.65 | 0.85 | 0.85 | 0.9 | 0.85 | 0.9 | 0.85 |
| ACMGCN-Moscat∗ | 0.55 | 0.65 | 0.65 | 0.85 | 0.9 | 0.85 | 0.9 | 0.9 |

# F Experimental Settings

## F.1 Datasets

We evaluate `Moscat` on datasets with various homophily ratios. We used the filtered version [53] of `Squirrel` datasets, which removes all the duplicated nodes that share the same neighbors and

Table 8: Base GNN model selected for `GNN-Moscat*` on each dataset in Table 2.

| | Squirrel | snap-patents | arxiv-year | Flickr | amazon-ratings | Penn94 | genius | ogbn-arxiv |
|---|---|---|---|---|---|---|---|---|
| Base GNN for GNN-Moscat(*) | GCNII | MixHop | GAT | GAT | ACMGCN | MixHop | GCNII | GAT |

Table 9: Comparison of `Moscat` with other methods use validation set for model training.

| Method | Squirrel | amazon-ratings | Penn94 | Method | Squirrel | amazon-ratings | Penn94 |
|---|---|---|---|---|---|---|---|
| GCN | $41.33 \pm 1.46$ | $48.55 \pm 0.38$ | $82.54 \pm 0.43$ | GAT | $39.36 \pm 1.89$ | $49.78 \pm 0.47$ | $81.81 \pm 0.62$ |
| GCN-ft | $41.36 \pm 2.49$ | $48.34 \pm 0.88$ | $83.07 \pm 0.32$ | GAT-ft | $39.74 \pm 2.21$ | $49.92 \pm 0.39$ | $82.65 \pm 0.75$ |
| GCN-mlp | $40.37 \pm 1.76$ | $48.79 \pm 0.57$ | $82.60 \pm 0.50$ | GAT-mlp | $38.96 \pm 2.25$ | $50.32 \pm 0.40$ | $81.87 \pm 0.60$ |
| GCN-Moscat* | $42.91 \pm 1.96$ | $51.02 \pm 0.50$ | $85.51 \pm 0.42$ | GAT-Moscat* | $43.10 \pm 1.98$ | $52.56 \pm 0.40$ | $85.41 \pm 0.48$ |
| GCN-Moscat | $43.55 \pm 2.08$ | $52.25 \pm 0.69$ | $85.76 \pm 0.32$ | GAT-Moscat | $43.10 \pm 2.13$ | $53.77 \pm 0.61$ | $85.35 \pm 0.59$ |

labels. These duplicates exist widely in the train and test set and can cause data leakage. The `amazon-ratings` dataset is from [53]. For the `Penn94`, `arxiv-year`, `genius` and `snap-patents` datasets, we follow the same settings as [35] with 50%/25%/25% random splits for train/valid/test. We run 10 times on each of the 10 benchmark datasets. We follow the setup in [53], which does not convert the directed graphs to undirected graphs and does not use reverse edges since the outgoing neighbors might not be observed during real-world inference.

## F.2 `Moscat*` and `Moscat`

**`Moscat*`.** In this setting, we reserve the original validation set $\mathcal{V}_{\text{val}}$ exclusively for validating both the GNN experts and the gating model, ensuring $\mathcal{V}_{\text{val}}$ is not used during model training. We introduce a hyperparameter $\alpha \in (0,1)$ to denote the ratio of data sampled from the original training set $\mathcal{V}_{\text{train}}$ for expert training $\mathcal{V}_{\text{exp}}$. The remaining $\mathcal{V}_{\text{train}}$ is designed as the holdout set $\mathcal{V}_{\text{hold}}$, reserved for gating training. To alleviate the experts' performance drop caused by training on only a subset of $\mathcal{V}_{\text{train}}$, we create a complementary split on $\mathcal{V}_{\text{train}}$ with the same $\alpha$. In this split, the expert-training set $\mathcal{V}'_{\text{exp}}$ completely overlaps with the holdout set $\mathcal{V}_{\text{hold}}$ from the first split. We then follow the same procedure to train another set of experts and gating models. At inference time, we average the outputs of the two gating models to form the final prediction. Adding the complementary split results in an average accuracy improvement of around 0.3, with also no validation set $\mathcal{V}_{\text{val}}$ used during training. In practice, `Moscat*` achieves its best results when $\alpha$ is between $0.55 \sim 0.9$ and remains stable across this range. The configuration of $\alpha$ is shown in Table 7.

**`Moscat`.** In this setting, we follow prior work [29, 62] to train the gating model on the original validation set. In our experiments, we sample 90% of the original validation set $\mathcal{V}_{\text{val}}$ as $\mathcal{V}_{\text{hold}}$ and use the remaining 10% for gating validation. In some cases we observe further gains by training the gating model on the full $\mathcal{V}_{\text{val}}$ and validating on $\mathcal{V}_{\text{train}}$. Thereby, we introduce a binary hyperparameter $\eta \in \{0, 0.1\}$, where $\eta = 0.1$ denotes the former split and $\eta = 0$ the latter.

Importantly, the validation set serves as a natural holdout set for assessing generalization across training epochs and hyperparameters. Our method, `Moscat`, can be seen as a fine-grained extension of this idea: rather than using $\mathcal{V}_{\text{val}}$ merely for early stopping, we leverage it to evaluate and weight different experts over node subgroups. In practice, `Moscat` is a plug-and-play module that operates on pre-trained GNN checkpoints without any retraining of the base model; we only train a lightweight MLP gating model on the modestly sized validation set (which seldom includes filtered samples from the training set), making it both fast and memory-efficient.

To verify that our performance gains arise from expert mixing rather than simply from extra training data, we compare against two baselines in Table 9 under the same validation split as `Moscat`:

- **GNN-ft:** fine-tuning the entire pre-trained GNN on $\mathcal{V}_{\text{val}}$.

- **GNN-mlp:** freezing the GNN backbone and training an auxiliary MLP classifier (taking GNN logits as input) on $\mathcal{V}_{\text{val}}$.

The results indicate that additional training on the validation set offers, at best, marginal accuracy improvements and sometimes leads to degradation compared to standard GNNs, likely due to the small size of $\mathcal{V}_{\text{val}}$ and the risk of overfitting. Together with the consistently superior gains of `Moscat*`,

Table 10: Hyperparameter settings for `Moscat`. We additionally include the original node features as the gating model input for all expert models in the `amazon-ratings` dataset. We observe that node features are highly informative for the gating model prediction on the `amazon-ratings` dataset, but they have negative effects or no impact on other datasets.

| Dataset | Model | `Moscat` specific hyperparameters | | | | Hyperparameters for MLP | | |
|---|---|---|---|---|---|---|---|---|
| | | $L_{max}$ | $\eta$ | $\gamma$ | mask wrong | learning rate | hidden dim. | norm |
| Squirrel | SGC-Moscat | 6 | 0.1 | - | ✓ | 0.001 | 128 | - |
| | GCN-Moscat | 5 | 0.1 | - | ✓ | 0.005 | 128 | - |
| | GAT-Moscat | 5 | 0.1 | - | ✓ | 0.001 | 128 | - |
| | GCNII-Moscat | 5 | 0.1 | - | ✓ | 0.0005 | 128 | - |
| | ACM-GCN-Moscat | 5 | 0.1 | - | - | 0.001 | 128 | - |
| amazon-ratings | SGC-Moscat | 6 | 0.1 | 1 | ✓ | 0.00005 | 256 | batch |
| | GCN-Moscat | 6 | 0.1 | 0.5 | - | 0.00005 | 256 | batch |
| | GAT-Moscat | 6 | 0.1 | 0.3 | ✓ | 0.0001 | 256 | batch |
| | GCNII-Moscat | 6 | 0.1 | 0.9 | ✓ | 0.0001 | 256 | batch |
| | ACM-GCN-Moscat | 6 | 0 | - | ✓ | 0.00005 | 256 | batch |
| Penn94 | SGC-Moscat | 6 | 0.1 | - | ✓ | 0.005 | 256 | batch |
| | GCN-Moscat | 6 | 0.1 | - | ✓ | 0.005 | 256 | batch |
| | GAT-Moscat | 6 | 0.1 | - | ✓ | 0.0001 | 256 | batch |
| | GCNII-Moscat | 6 | 0.1 | - | - | 0.005 | 256 | batch |
| | ACM-GCN-Moscat | 6 | 0.1 | - | ✓ | 0.00005 | 256 | layer |
| Flickr | SGC-Moscat | 6 | 0 | - | - | 0.001 | 256 | - |
| | GCN-Moscat | 6 | 0 | - | - | 0.001 | 256 | - |
| | GAT-Moscat | 6 | 0 | - | ✓ | 0.001 | 256 | - |
| | GCNII-Moscat | 6 | 0.1 | - | - | 0.0001 | 256 | - |
| | ACM-GCN-Moscat | 6 | 0.1 | - | - | 0.001 | 256 | - |
| arxiv-year | SGC-Moscat | 6 | 0 | - | - | 0.005 | 256 | layer |
| | GCN-Moscat | 6 | 0.1 | - | - | 0.005 | 256 | layer |
| | GAT-Moscat | 6 | 0 | - | - | 0.001 | 256 | layer |
| | GCNII-Moscat | 6 | 0.1 | - | - | 0.005 | 256 | layer |
| | ACM-GCN-Moscat | 6 | 0 | - | - | 0.005 | 256 | layer |
| genius | SGC-Moscat | 6 | 0.1 | 0 | - | 0.0005 | 256 | batch |
| | GCN-Moscat | 6 | 0.1 | 0 | - | 0.00005 | 256 | batch |
| | GAT-Moscat | 6 | 0.1 | 0 | - | 0.00005 | 256 | layer |
| | GCNII-Moscat | 6 | 0.1 | 0 | - | 0.00005 | 256 | layer |
| | ACM-GCN-Moscat | 6 | 0.1 | 0 | - | 0.00005 | 256 | layer |
| snap-patents | SGC-Moscat | 6 | 0 | - | - | 0.005 | 128 | batch |
| | GCN-Moscat | 6 | 0 | - | - | 0.005 | 128 | batch |
| | GAT-Moscat | 6 | 0 | - | - | 0.005 | 128 | batch |
| | GCNII-Moscat | 6 | 0 | - | - | 0.005 | 128 | batch |
| | ACM-GCN-Moscat | 6 | 0 | - | - | 0.005 | 128 | batch |
| ogbn-arxiv | SGC-Moscat | 6 | 0.1 | 1 | - | 0.001 | 256 | layer |
| | GCN-Moscat | 6 | 0.1 | 0.9 | - | 0.0005 | 256 | layer |
| | GAT-Moscat | 6 | 0.1 | 0.9 | ✓ | 0.0005 | 256 | layer |
| | GCNII-Moscat | 6 | 0.1 | 1 | - | 0.0005 | 256 | layer |
| | ACM-GCN-Moscat | 6 | 0.1 | 1 | ✓ | 0.0005 | 256 | layer |

these results confirm that our improvements stem from adaptive expert mixing rather than from naively adding more labeled training data.

## F.3 Baselines

To evaluate the flexibility of `Moscat`, we select three classic homophilous GNNs (GCN [28], SGC [65], GAT [60]), and two state-of-the-art heterophilous GNNs (GCNII [8], ACM-GCN [40]) which cover comprehensive GNN architectural designs. We note that many scope mixing methods have assumptions about GNN architectures. For example, [18] can only be used on decoupled GNNs like SGC, while [40, 34, 70] are incompatible with GNNs using learnable aggregators like GAT. We further compare our methods with four models designed for node classification under heterophily: H2GCN [83], GPRGCN [11], FSGNN [46], MixHop [1]; three Graph Transformers: GraphGPS [55], SGFormer [66], Polynormer [15]; and three Graph MoEs: GMoE [61], Mowst [78], DA-MoE [75]; We also include skip-connection methods designed for deeper GNNs: Jumping Knowledge [68], GAMLP [81] and $G^2$-GNN [57].

## F.4 Hyperparameters

In this section, we first clarify the hyperparameter settings for both GNNs and `Moscat`. Then, we provide detailed instructions and analysis for `Moscat` tuning.

Table 5 shows the hyperparameter search range for baseline GNNs. We set them according to the original paper for other special hyperparameters. Unless stated, we do not tune GNNs with residual connections or jumping knowledge. We also include an extra dropout layer on the input node feature, using a dropout ratio different from the one after each hidden layer. For SGC, we use an MLP instead of a single linear layer. For every MLP, including the one used in `Moscat`, we add residual connections and fix the number of layers to 3.

We observe that normalization is important for GNNs, especially on larger datasets, and tuning the depth can provide substantial accuracy gains. With proper hyperparameter tuning, classic GNNs are strong baselines on heterophilous graphs, which aligns with recent works' observation [53, 42]. When tuning the hyperparameters of the MLP in `Moscat`, we notice it is not sensitive to the number of transformation layers. Setting the number of layers to 3 works well in every setting. We also find that `Moscat` faces an accuracy drop when setting dropout or feature dropout to a value larger than 0. Therefore, we set the layers of MLP to 3 and dropout to zero for all settings, as shown in Table 10.

We further investigate tuning the `Moscat`-specific hyperparameters. Table 6 displays each hyperparameter's explanation and search range. For the hidden dimension of `Moscat`, we notice that `Moscat` often works well when setting the dimension to 256. In smaller datasets such as `Chameleon`, we observe that a lower dimension is enough and enjoys better training. We limit the dimension for the `snap-patents` dataset to 128 due to the GPU memory constraint. Note that we set the upper bound of the `Moscat` maximum scope size $L_{max}$ to 6, rather than limiting it to a shallow 2 to 3 hop neighborhood [77] or expanding it to a global scope [33]. This is because (1) shallow scope does not contain sufficient homophily in heterophilous graphs [83], and (2) according to the six degrees of separation theorem, a 6-hop neighborhood is already large enough for many Wikipedia, citation, and social networks. Further increasing the scope size will only provide marginal improvement. As shown in Table 10, setting $L_{max}$ to 6 works best in most cases. $L_{max}$ can be used as a crucial parameter to trade off accuracy and overhead (see Appendix G.4 for more details).

Table 10 shows the best hyperparameters for `Moscat` results presented in Table 1. `Moscat`∗ uses the same hyperparameter configuration as `Moscat`, with an additional hyperparameter $\alpha$ tuned separately (see Table 7 for best configuration and Appendix F.2 for more details). We additionally include the original node features as one of the inputs for all models in the `amazon-ratings` dataset (not for other datasets) since we observe that node features are highly informative for `amazon-ratings` but have adverse or negligible effects on other datasets.

## F.5 Software and Hardware

We implement `Moscat` using Python 3.11, PyTorch 2.0.1, PyG 2.4.0, and CUDA 12.2. All the experiments are conducted on a machine with dual 96 Core AMD EPYC 9654 CPUs paired with

Table 11: Leaderboard comparison. ACM-GCN++ and GloGNN++ use MLPs to transform the entire adjacency matrix, which is only applicable to the transductive setting.

| Rank | amazon-ratings | Penn94 | PubMed |
|---|---|---|---|
| 1st | 55.54 ±0.51 
 tuned-GAT [43] | 86.18 ±0.24 
 PathMLP [82] | 91.95 ±0.19 
 GNNDLD [6] |
| 2nd | 54.92 ±0.42 
 NID [42] | 86.09 ±0.56 
 Dual-Net GNN [47] | 91.56 ±0.50 
 NHGCN [21] |
| 3rd | 54.81 ±0.49 
 Polynormer [15] | 86.08 ±0.43 
 ACM-GCN++ [40] | 91.44 ±0.59 
 ACM-Snowball-3 [40] |
| Ours | **56.66** ±0.57 
 SAGE-Moscat ($L_{max} = 6$) | **87.13** ±0.45 
 MixHop-Moscat ($L_{max} = 16$) | **92.40** ±0.46 
 GCNII-Moscat ($L_{max} = 6$) |

1.5TB ECC-DDR5 RAM and a single NVIDIA RTX 6000 Ada GPU with 48GB ECC-GDDR6 VRAM.

# G Additional Experiments and Analysis

## G.1 Leaderboard Comparison

Since `Moscat` is flexible for GNNs with various architectures, can `GNN-Moscats` achieve state-of-the-art performance? To answer this, Table 11 summarizes our comparison with the top 3 methods reported on `amazon-ratings`, `Penn94`, and `PubMed` leaderboards from Paper With Code. To the best of our knowledge, we find two methods, NID [42] and PathMLP [82], have reported top performance but have not been included in the leaderboard. We also include this method in our comparison. The result demonstrates the superior performance of `GNN-Moscat`. Note that we follow the standard setup for training the base GNN models and do not include any additional tricks.

## G.2 Runtime Analysis

Let $L'$ denote the number of MLP layers and $F_{hid}$ denote the hidden dimension. The time complexity of the gating model $\mathcal{F}$ is bounded by $O\left(CL_{max}|\mathcal{V}|F_{hid} + |\mathcal{V}|L_{max}^2 + L'|\mathcal{V}|F_{hid}^2\right)$.

Although `GNN-Moscat` requires first individually training $L_{max} + 1$ GNN experts and then training an additional gating model, it does not add much overhead during training. This is due to:

1. Model depth is one of the most critical hyperparameters for GNNs on heterophilous graphs, with deeper models typically achieving better performance. Regardless, training GNNs with 1 to 6 layers is necessary during the hyperparameter search for the optimal number of layers L. Table 12 shows the accuracy when fixing the number of layers of the baseline GNNs to 2, as well as the accuracy when selecting the optimal number of layers through hyperparameter tuning.

2. Since GNNs of different depths can be trained in parallel, the total (parallel) training time for `GNN-Moscat` has two components: the time to train the deepest GNN ($L = L_{max}$) and the gating model training time. The simplicity of our design enables efficient utilization of the GPU resources, while other personalized scoping methods require complex architecture (e.g., GNN-G$^2$ employs additional GNNs for gating at each layer) or significantly longer training epochs (e.g., Mowst iteratively trains GNNs and the gating module). Table 13 shows the parallel training time of `Moscat` remains low compared to other methods.

3. `Moscat` can be flexibly applied to shallow GNNs only while maintaining significant accuracy improvements, reducing the computational overhead of training deeper GNNs. As shown in Figure 7 and 8, the majority of accuracy gains from `Moscat` occur with GNNs of depth $L \leq 4$.

Table 12: Performance comparison for GNNs with different depths.

| Model | | Chameleon | Squirrel | arxiv-year | genius |
|---|---|---|---|---|---|
| SGC | L=2 | 38.79 | 39.55 | 45.28 | 87.53 |
| | L=best | 39.72 (L=5) | 40.04 (L=1) | 45.88 (L=4) | 88.01 (L=1) |
| GCN | L=2 | 39.75 | 40.56 | 43.76 | 89.15 |
| | L=best | 41.74 (L=6) | 41.33 (L=6) | 48.68 (L=5) | 90.22 (L=4) |
| GAT | L=2 | 37.97 | 37.57 | 50.16 | 86.53 |
| | L=best | 39.93 (L=5) | 39.36 (L=4) | 52.77 (L=3) | 88.44 (L=4) |

Table 13: Parallel training time comparison.

| | Chameleon | amazon-ratings | Penn94 |
|---|---|---|---|
| GCN (L=6) | 1.59s ($\times 1.00$) | 12.67s ($\times 1.00$) | 7.27s ($\times 1.00$) |
| Moscat | $\times 0.19$ | $\times 0.16$ | $\times 0.23$ |
| GCN-Moscat | $\times 1.19$ | $\times 1.16$ | $\times 1.23$ |
| GCN-Attn | $\times 1.40$ | $\times 2.63$ | $\times 2.34$ |
| GCN-G$^2$ | $\times 1.79$ | $\times 13.6$ | $\times 3.36$ |
| GCN-Mowst | $\times 61.6$ | $\times 41.9$ | $\times 23.4$ |

## G.3 Space Analysis

Moscat allows each expert and the gating model to be trained separately. Practitioners can flexibly adjust the parallelism of expert training to balance runtime and memory consumption. Let $L_{\max}$ denote the maximum depth of GNN experts, $F_{\text{hid}}$ denote the hidden dimension, and $\mathcal{V}_{\text{exp}}$ and $\mathcal{V}_{\text{gate}}$ denote the training sets for experts and the gating model, respectively. Taking GCN as an example, we have two boundary cases:

- If we prioritize minimal memory overhead, we can train each expert sequentially. The space complexity is therefore bounded by training the deepest GNN expert: $O\big(L_{\max}|\mathcal{V}_{\text{exp}}|F_{\text{hid}} + L_{\max}F_{\text{hid}}^2\big)$.
- If we prioritize minimal runtime, we can train all experts in parallel. The space complexity then becomes proportional to the total number of layers $(1 + 2 + \cdots + L_{\max})$ across all experts: $O\big(L_{\max}^2|\mathcal{V}_{\text{exp}}|F_{\text{hid}} + L_{\max}^2F_{\text{hid}}^2\big)$.

The memory consumption of the gating model training is relatively small and won't become a bottleneck: $O\big(L_{\max}|\mathcal{V}_{\text{gate}}|F_{\text{hid}} + L_{\max}F_{\text{hid}}^2\big)$, since $F_{\text{hid}} \ll |\mathcal{V}_{\text{gate}}| \ll |\mathcal{V}_{\text{exp}}|$. In conclusion, with comparable runtime, Moscat requires approximately $L_{\max}$ times more memory than a single $L_{\max}$-layer GNN during training.

## G.4 Depth Analysis

We investigate how GNN-Moscat performance changes when varying the maximum depth $L_{\max}$ of GNN experts. Figure 7 and Figure 8 compare the performance of GNN-Moscat and corresponding GNN on amazon-ratings dataset and arxiv-year dataset, respectively. As depth increases, GNN performance first increases and then decreases or remains unchanged, which is undesirable given deeper GNNs' superior expressive power [13]. In contrast, GNN-Moscat leverages depth more effectively, showing a consistent upward trend in performance as expert depth increases. This indicates a promising characteristic of GNN-Moscat: it does not require careful tuning of $L_{\max}$ tuning to achieve good performance. We found that $L_{\max} = 6$ is a good trade-off between accuracy and computational overhead for all datasets and all base GNNs. For cases with strict runtime requirements, setting $L_{\max} = 2$ for GNN-Moscat consistently achieves performance that is better than or comparable to the best GNN with depths between 1 and 10.

## G.5 Empirical Evidence of Deeper Soft Scoping Methods Suffer from Overfitting

We plot the training and test accuracy of a representative soft scoping method ACMGCN [40] (see Figure 9). The figure demonstrates that the 6-layer ACMGCN achieves higher training accuracy than the 2-layer version. However, in homophily regions, the 6-layer model exhibits lower test accuracy. This suggests that while deeper soft scoping methods offer greater expressive power, they are also

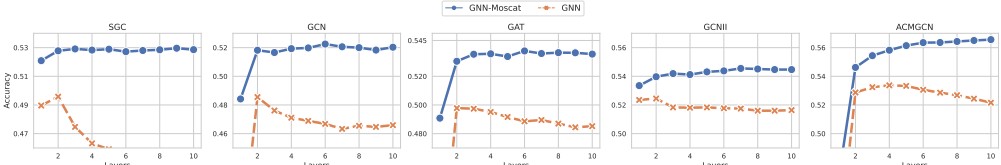

Figure 7: Performance comparison between GNN and `GNN-Moscat` on `amazon-ratings` dataset. The x-axis represents the number of layers for GNN and $L_{\max}$ for `GNN-Moscat`.

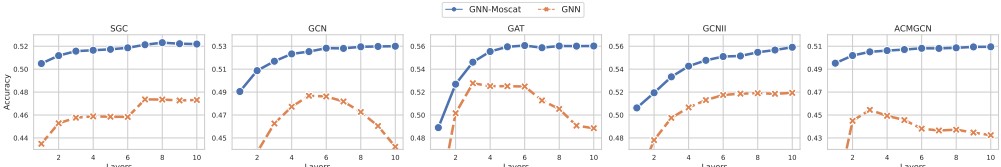

Figure 8: Performance comparison between GNN and `GNN-Moscat` on `arxiv-year` dataset. The x-axis represents the number of layers for GNN and $L_{\max}$ for `GNN-Moscat`.

prone to overfitting. Considering the architecture of soft scoping methods, we speculate this issue arises because shallow layers/experts in soft scoping models may overfit to noise from higher-hop neighbors.

### G.6 Supplementary Details for Heterophily-Biased Sample Filtering

Heterophily nodes exhibit more diverse neighborhood label patterns than homophily nodes, which makes them more challenging for experts to generalize from. As a result, experts typically show a larger generalization gap on heterophily nodes. When experts overfit $\mathcal{V}_{\text{exp-het}}$ (i.e., the heterophilous nodes in $\mathcal{V}_{\text{exp}}$), their logits on these samples show high prediction accuracy with high confidence. Consequently, the gating model may incorrectly assign large weights to these overfitted experts on heterophily nodes after training on $\mathcal{V}_{\text{exp-het}}$.

To further illustrate the effectiveness and reasoning behind our $\mathcal{V}_{\text{exp-het}}$ filtering technique, we demonstrate an example using GCN and SGC on the `amazon-ratings` dataset (see Figure 10). When comparing the homophily-accuracy curves for experts of depths 0 to 6 across the training, validation, and test sets, we observe that:

- The curves for the validation and test sets are nearly identical.
- Although the training set's curves align with those on the validation set in the homophily region, they show significant deviations in the heterophily region.

These findings motivate us to exclude samples from $\mathcal{V}_{\text{exp-het}}$ when training the gating model.

### G.7 How Expert Training Affects `Moscat` Performance

In this section, we analyze how expert training affects performance by varying the dropout ratio, hidden dimensions, and training epochs for each expert. We set $\gamma$ to "-" for all `GNN-Moscat` models in this analysis. Although we did not tune the GNN experts for enhanced `GNN-Moscat` performance in Table 2, the insights from this hyperparameter tuning suggest that further improvements in accuracy can be achieved for `GNN-Moscat` models.

Table 14 presents a performance comparison between GAT and GCNII, along with their `Moscat` variants (`GAT-Moscat` and `GCNII-Moscat`), across various dropout ratios on two homophilous datasets: `arxiv-year` and `PubMed`. We observe that as the dropout ratio decreases, the accuracy of both GAT and GCNII experiences a slight decline, suggesting an increased risk of overfitting. In contrast, the `Moscat` versions exhibit an improving accuracy trend.

This phenomenon is further explained by analyzing the distribution of nodes into three categories:

1. "All wrong" — nodes misclassified by all experts.

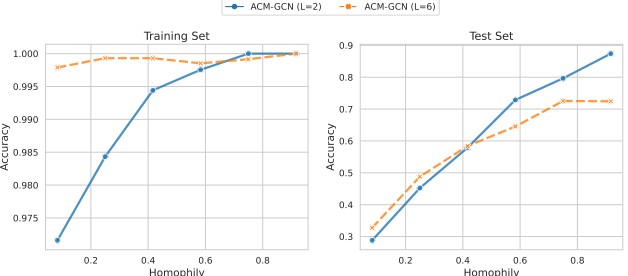

Figure 9: Training and test accuracy of the shallow ($L = 2$) and deeper ($L = 6$) soft scoping method ACMGCN on the `amazon-ratings` dataset. Results indicate that deeper ACMGCN exhibits more pronounced overfitting, particularly on homophilous nodes.

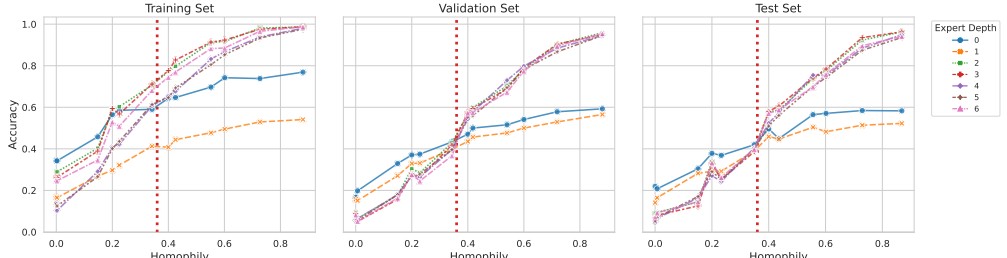

Figure 10: The accuracy of **GCN** experts on training/validation/test sets of the `amazon-ratings` dataset. The red dotted line shows the average homophily ratio of the training set.

2. "All correct" — nodes correctly classified by all experts.

3. "Others" — nodes that do not fall into the above two categories

Table 14 indicates that with decreasing dropout, the percentages of "all wrong" and "all correct" nodes both decrease, while the proportion of "others" increased. This shift creates a larger margin for possible performance gains when applying `Moscat`. Moreover, both `GAT-Moscat` and `GCNII-Moscat` consistently outperform their standard GNN counterparts across different dropout levels. This suggests that incorporating `Moscat` enhances robustness and stability across varying dropout settings.

We further compare the performance of GCN, GAT, and their `Moscat` variants on the heterophilous dataset `amazon-ratings`. Table 15 illustrates how test and training accuracies evolve as the hidden dimensions for all experts increase. For both GCN and GAT, test accuracy shows only modest improvements while training accuracy increases substantially, highlighting an increased generalization gap. In contrast, although the test accuracies of `GCN-Moscat` and `GAT-Moscat` also improve, the performance gains relative to their base models follow different trends. Specifically, the accuracy gains of `GCN-Moscat` over GCN gradually decrease with larger hidden dimensions, whereas those of `GAT-Moscat` over GAT continue to increase. This divergence is explained by the observation that, with increasing hidden dimensions, the percentage of "all correct" for GCN experts rises, while the percentage of "all wrong" for GAT experts falls—leading to distinct variations in the performance margins when applying `Moscat`.

To investigate the effect of expert training on `Moscat` performance, we selected experts from checkpoints at 300, 1000, and 2000 training epochs, corresponding to underfitting (i.e., low test and training accuracy), well-fitting (i.e., good test and training accuracy), and overfitting scenarios (i.e., similar test accuracy but much higher training accuracy), respectively. Figure 16 shows that as training epochs increase, for both GCN and GAT experts, the frequency of "all wrong" predictions decreases, while "all correct" predictions initially rise and then decline. For `GNN-Moscat`, it is desirable for experts to fit the training data properly; however, the benefit of overfitting depends on the expert architecture (e.g., GAT benefits more from overfitting, whereas GCN does not).

Overall, these experiments suggest promising directions for tuning experts to enhance `Moscat` performance further. Additionally, we note that GNN experts exhibit a higher percentage of "others"

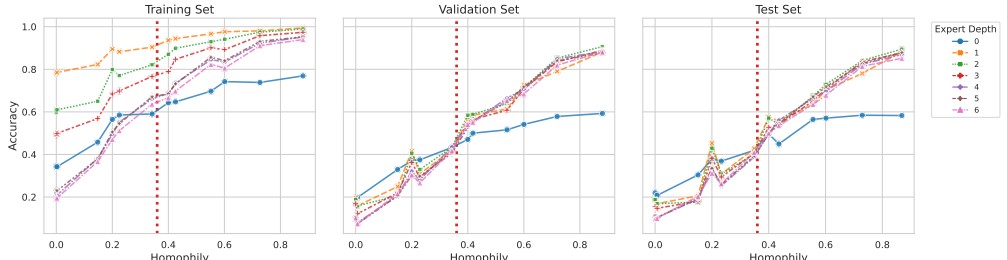

Figure 11: The accuracy of **SGC** experts on training/validation/test sets of the `amazon-ratings` dataset. The red dotted line shows the average homophily ratio of the training set.

Table 14: Test accuracy comparisons across different dropout ratios for GNN experts. We set $L_{\max} = 6$ and fix other GNNs and `Moscat` hyperparameters. For GAT and GCNII, we report their best accuracy across configurations with 1-6 layers.

| | Dropout | 0.5 | 0.3 | 0.1 |
|---|---|---|---|---|
| | GAT | 72.14 ±0.17 | 72.17 ±0.17 | 72.01 ±0.22 |
| `ogbn-arxiv` | All wrong | 18.4% | 18.0% | 17.6% |
| | All correct | 56.5% | 56.5% | 56.0% |
| | Others | 25.1% | 25.6% | 26.4% |
| | GAT-Moscat | 73.69 ±0.19 | 73.85 ±0.20 | 73.91 ±0.26 |
| | Δ | +1.55 | +1.68 | +1.90 |
| | GCNII | 91.27 ±0.51 | 91.18 ±0.72 | 91.20 ±0.65 |
| `PubMed` | All wrong | 5.4% | 4.6% | 4.6% |
| | All correct | 86.1% | 84.8% | 84.7% |
| | Others | 8.5% | 10.6% | 10.7% |
| | GCNII-Moscat | 92.04 ±0.33 | 92.32 ±0.34 | 92.24 ±0.37 |
| | Δ | +0.83 | +1.14 | +1.04 |

Table 15: Test accuracy comparisons across different hidden dimensions for GNN experts. We set $L_{\max} = 6$ and fix other GNNs and `Moscat` hyperparameters. For GCN and GAT, we report their best accuracy across configurations with 1-6 layers. (00.00 ±0.00) denotes the corresponding training accuracy.

| | Hidden Dimensions | 128 | 256 | 512 |
|---|---|---|---|---|
| | GCN | 47.33 ±0.52 (59.03 ±1.54) | 48.18 ±0.66 (61.51 ±2.33) | 48.55 ±0.38 (63.25 ±2.85) |
| | All wrong | 30.1% | 30.1% | 30.0% |
| | All correct | 16.8% | 17.2% | 17.4% |
| | Others | 53.1% | 52.7% | 52.6% |
| `amazon-ratings` | GCN-Moscat | 50.75 ±0.87 | 51.07 ±0.50 | 51.34 ±0.68 |
| | Δ | +3.42 | +2.89 | +2.79 |
| | GAT | 48.55 ±0.51 (59.19 ±1.56) | 49.42 ±0.56 (65.20 ±0.88) | 49.78 ±0.47 (72.42 ±0.99) |
| | All wrong | 29.5% | 28.8% | 27.0% |
| | All correct | 17.2% | 17.2% | 17.0% |
| | Others | 53.3% | 54.0% | 56.0% |
| | GAT-Moscat | 50.68 ±0.57 | 51.57 ±0.53 | 52.43 ±0.57 |
| | Δ | +2.13 | +2.15 | +2.65 |

on heterophilous graphs than on homophilous graphs, which may explain why `Moscat` achieves more significant accuracy improvements on heterophilous graphs.

### G.8 Interpretation and Visualization

The above experiments have demonstrated the effectiveness of `Moscat` in improving GNN generalization by mixing GNN models with different scopes. To further understand how expert mixing influences generalization, we introduce a variant of `Moscat` called Adaptive Scope (`AS`). In this variant, instead of predicting node labels, the gating model predicts the IDs of scope experts.

Table 16: Test accuracy comparisons across different training epochs for GNN experts. We select checkpoints at 300, 1000, and 2000 training epochs for each expert, representing underfitting, well-fitting, and overfitting states, respectively. We set $L_{\max} = 6$, expert dropout to 0, and fix other GNNs and `Moscat` hyperparameters. For GCN and GAT, we report their best accuracy across configurations with 1-6 layers. (00.00 ±0.00) denotes the corresponding training accuracy.

| Training Epochs | | 300 (underfitting) | 1000 (well-fitting) | 2000 (overfitting) |
|---|---|---|---|---|
| | GCN | 44.73 ±0.90 (51.50 ±1.89) | 48.34 ±0.49 (65.47 ±0.92) | 48.22 ±0.63 (68.42 ±1.13) |
| | All wrong | 29.8% | 29.7% | 26.9% |
| | All correct | 11.7% | 17.9% | 16.0% |
| | Others | 58.6% | 52.4% | 57.1% |
| amazon-ratings | GCN-Moscat Δ | 49.03 ±0.72 +4.30 | 51.11 ±0.64 +2.77 | 51.27 ±0.74 +3.05 |
| | GAT | 46.17 ±0.57 (54.56 ±1.57) | 49.53 ±0.58 (67.44 ±0.91) | 49.59 ±0.97 (78.39 ±1.15) |
| | All wrong | 29.6% | 27.0% | 23.4% |
| | All correct | 12.9% | 16.8% | 16.4% |
| | Others | 57.5% | 56.2% | 60.2% |
| | GAT-Moscat Δ | 48.49 ±0.75 +2.32 | 52.12 ±0.64 +2.59 | 53.34 ±0.45 +3.75 |

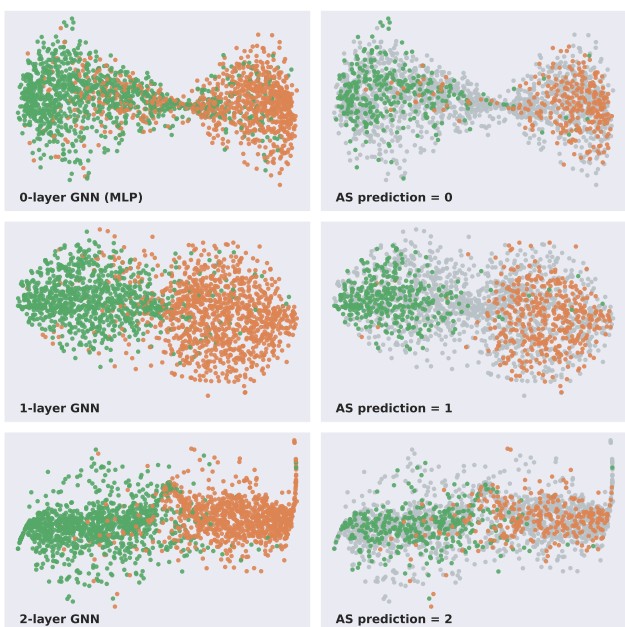

Figure 12: (Left) Visualization of t-SNE on GNN embeddings for `Penn94`. Green and orange dots indicate nodes with different labels. (Right) In each embedding visualization, nodes that AS (a variant of `Moscat`) predicts to have the corresponding depth are highlighted.

To analyze the behavior of AS, we use t-SNE [59] to visualize the hidden embeddings from three scope experts, which are GAT models with layers ranging from 0 to 2, on the `Penn94` dataset (Figure 12 Left). Specifically, we add an output linear layer to each GAT model and visualize the hidden embeddings just before this layer. Since AS predicts the optimal scope expert (i.e., Scope-0, Scope-1, or Scope-2) for each node, we highlight nodes according to the predicted scope expert in the embedding visualization (Figure 12 Right).

From Figure 12, we make three key observations:

1. AS tends to select nodes that the model can differentiate more easily.
2. The nodes chosen by AS are located near the center of each cluster.
3. An outlier in one model's embedding can serve as the center in another model's embedding.

These findings support our hypothesis that GNNs with different depths can generalize better on different subsets of nodes.

## G.9   More Empirical Findings of Performance Disparity across Scope Experts

Section 3.3 empirically examines our theoretical findings through several experiments. In this section, we extend these experiments to include more GNN models and datasets.

### G.9.1   Overlapping Ratio

Figure 13 and Figure 14 present heatmaps of the Jaccard overlap ratios (of correctly predicted test nodes) between pairs of scope experts (scope sizes 0–6) for five GNN architectures (SGC, GCN, GAT, GCNII, and ACMGCN) across multiple datasets. Darker cells denote lower overlap in each matrix, indicating that the corresponding expert pair correctly predicts largely distinct sets of test nodes, while lighter cells indicate higher overlap. Based on these two figures, we make several observations:

1. **Scope-0 and Scope-1 experts diverge most:** Across architectures and datasets, experts with scopes of 0 and 1 exhibit the lowest overlap with other experts, suggesting they capture unique information relative to larger-scope variants.

2. **Dataset homophily drives overlap:** Heterophilous datasets (e.g., `Chameleon`, `Squirrel`, and `snap-patents` with average node homophily less than 0.2) show consistently low overlaps (often < 0.7), whereas homophilous datasets (e.g., `arxiv-year`, `PubMed`) yield high overlaps (often > 0.7).

3. **Architecture-dependent diversity:** Attention-based models (GAT, ACMGCN) produce lower overlaps and more varied patterns, particularly GAT on Penn94 and ACMGCN on Pubmed, indicating greater specialization among experts. In contrast, the decoupled SGC architecture exhibits uniformly higher overlaps, reflecting more redundant predictions among its experts.

### G.9.2   Performance Disparity

Theorem 3.3 reveals a clear generalization disparity between shallow and deeper experts across node homophily subgroups. Specifically, the disparity is related to the average node homophily ratio in the training set. Although our theoretical analysis is based on SGC, our experiments indicate that the observations also hold for other, more complex GNNs.

Figure 15 illustrates the performance gap between deeper and shallow experts across several datasets. In the figure, a positive bar indicates that the deeper experts outperform the shallow ones. The results confirm our theory by showing significant performance differences between regions divided by the average homophily ratio.

We also notice distinct patterns across various GNN architectures:

- SGC and GCN: In regions with low homophily (heterophilous regions), deeper experts always perform worse than shallow experts.

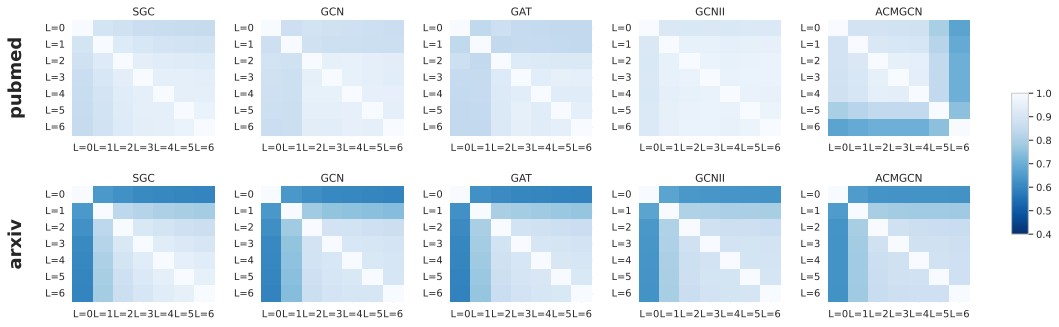

Figure 13: The overlapping ratio matrices for scope experts on homophilous datasets.

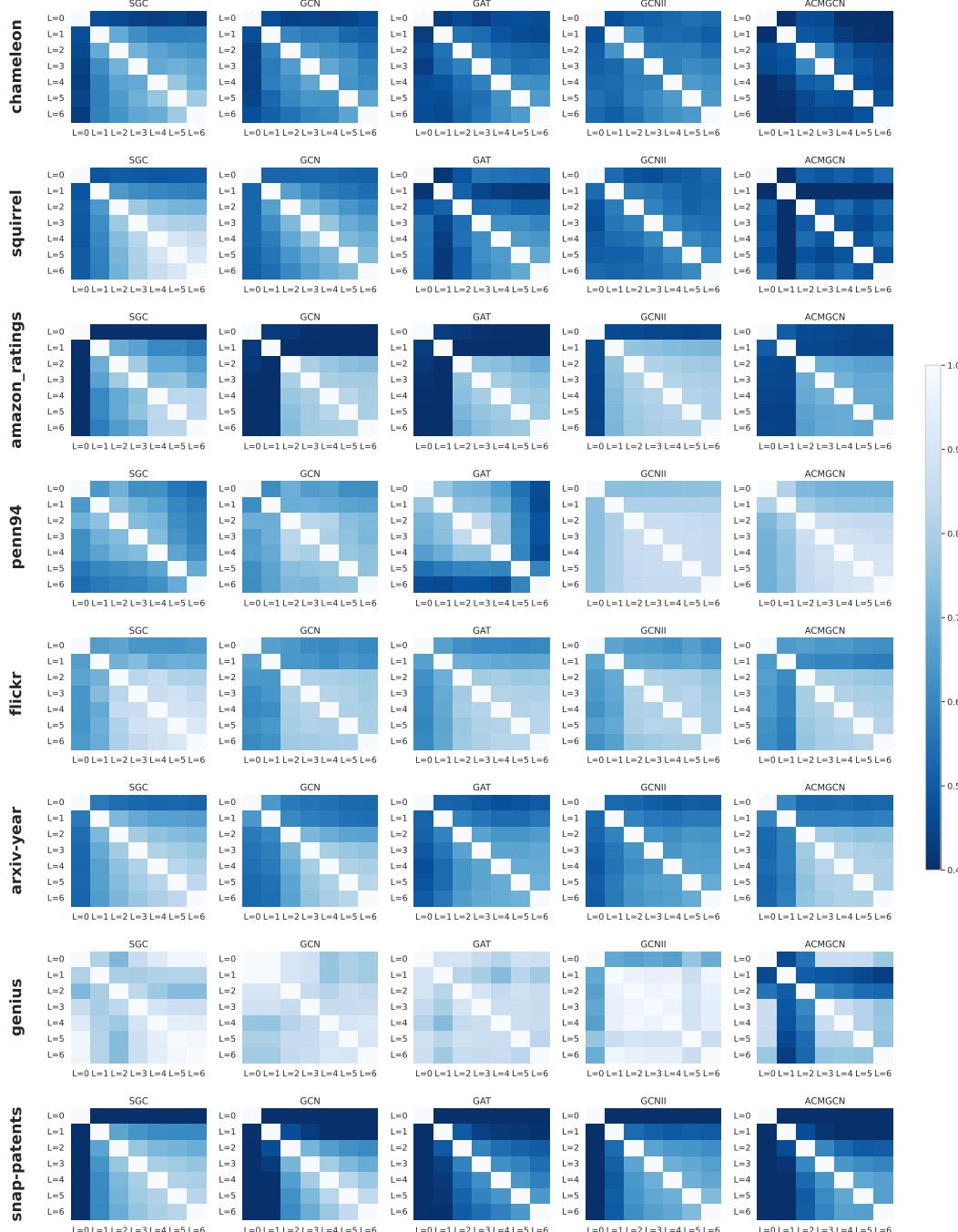

Figure 14: The overlapping ratio matrices for scope experts on heterophilous datasets.

- GAT and GCNII: In these architectures, deeper experts sometimes outperform shallow experts even in heterophilous regions.

- ACMGCN: Here, deeper experts consistently outperform shallow experts in heterophilous regions.

A possible explanation for these differences is that GAT, GCNII, and ACMGCN incorporate specific designs for handling heterophilous information. For instance, GAT uses attention-based neighbor aggregation, GCNII applies inception residual connections, and ACMGCN integrates both high-pass and full-pass filters. In contrast, SGC and GCN lack these mechanisms. In particular, stacking multiple high-pass and full-pass filters appears to be highly effective for learning heterophily.

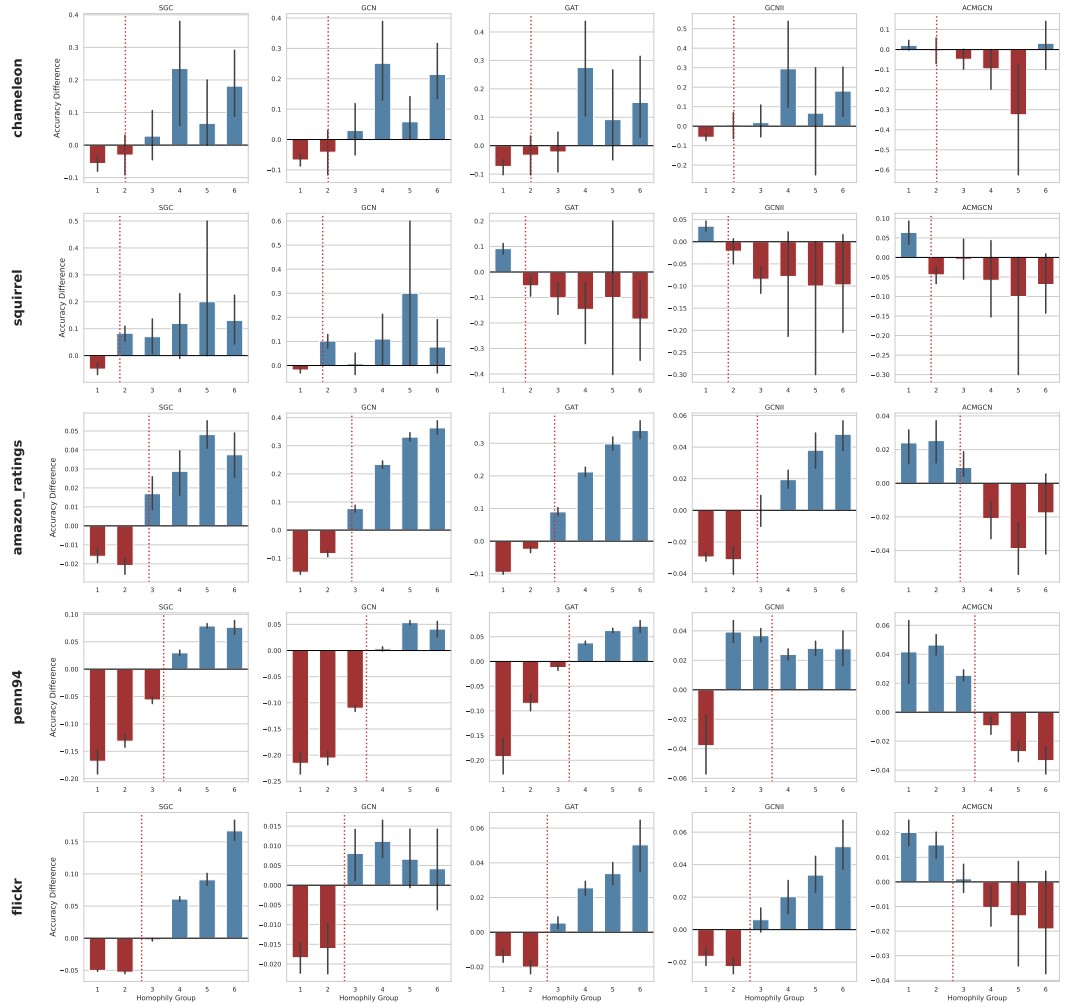

Figure 15: Test accuracy differences between deeper and shallow experts. Positive values (blue) indicate deeper experts perform better, while negative values show shallow experts perform better. The red dotted line shows the average homophily ratio of the training set. We have can observe the following trends. The deeper variants of homophilous GNNs (e.g., SGC, GCN) tend to perform worse than their shallow counterparts in heterophily regions. For GNNs designed for heterophily (e.g., GAT, GCNII, ACMGCN), in some cases, their deeper variants can outperform shallow ones in heterophily regions. Notably, the deeper variant of ACMGCN consistently outperforms its shallow variant across all datasets, which underscores the effectiveness of its high-pass filter in leveraging heterophily.

### G.9.3 Ensembling Upper-bound

We further investigate the extent of the observed generalization disparity across different GNN architectures and datasets. Specifically, for a given set of scope experts, we determine the upper bound by calculating the percentage of nodes correctly predicted by at least one expert. Figures 16 and 17 illustrate the upper bounds for homophilous and heterophilous datasets, respectively.

Across representative GNN architectures, all datasets exhibit sub-linear curves and notable accuracy improvements, ranging from approximately 10% to 50%, as $L_{\max}$ increases. This trend suggests that test-time scaling is promising by increasing the number of experts for different scopes.

Furthermore, our findings indicate that GAT, GCNII, and ACMGCN are best performing architectures with sufficiently large $L_{\max}$, whereas SGC tends to perform the worst. These results underscore the critical role of expert architectural design.

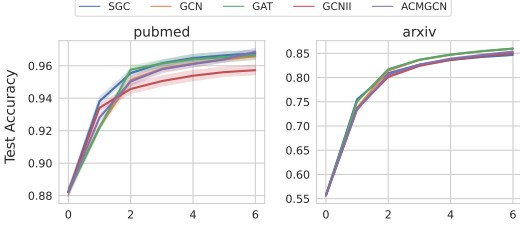

Figure 16: Upper-bound test accuracy achieved by scope expert ensemble on homophilous datasets. We report the percentage of nodes correctly predicted by at least one expert with the depth ranging from $L = 0$ to $n$ (where $n \leq 6$).

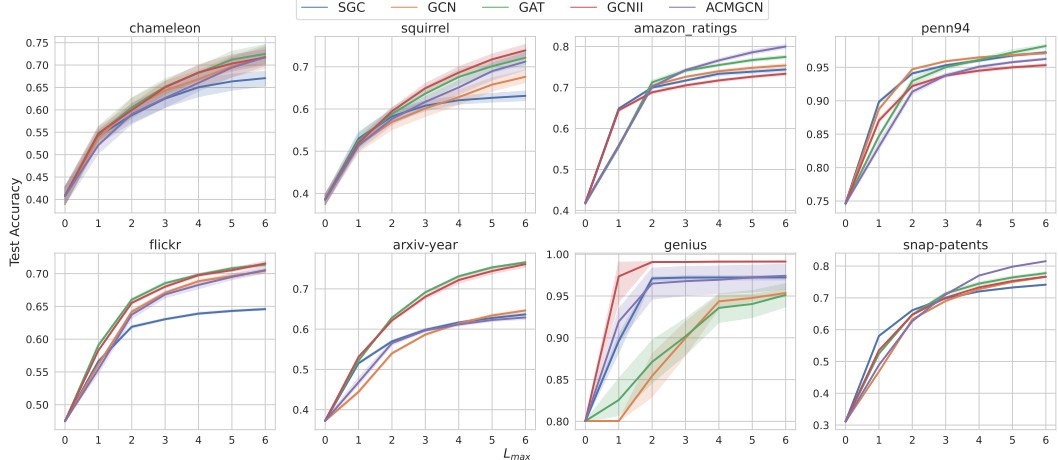

Figure 17: Upper-bound test accuracy achieved by scope expert ensemble on heterophilous datasets. We report the percentage of nodes correctly predicted by at least one expert with the depth ranging from $L = 0$ to $n$ (where $n \leq 6$).

### G.10    Analysis of the Gating Weights

Figure 18 and Figure 19 illustrate the distributions of gating weights for various experts in `Moscat`. The results indicate that `Moscat` effectively learns gating weights, thereby preventing the collapse issue (i.e., some experts receive extremely small weights across nodes) as noted in prior studies [55, 58]. Moreover, the gating model tends to assign larger weights with greater variance to shallow experts (e.g., Scope-0 and Scope-1) compared to deeper ones. This behavior suggests that `Moscat`

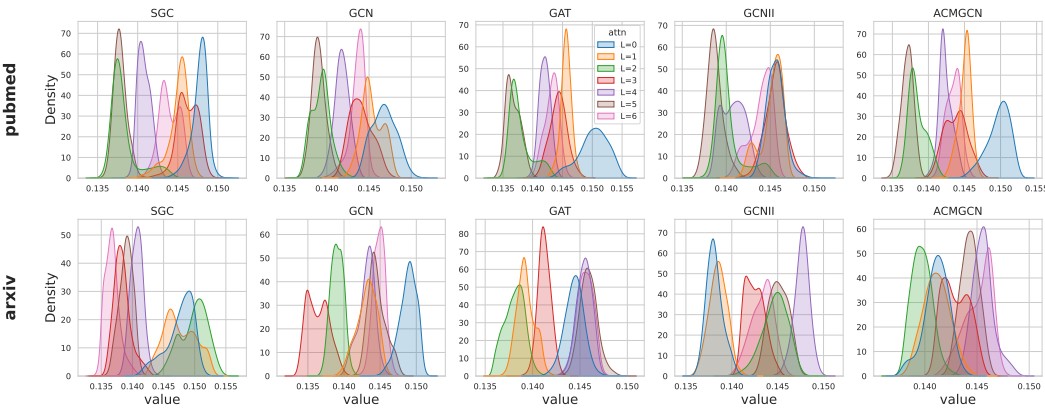

Figure 18: Learned attention-based gating weight distributions on homophilous datasets. Different color denotes the weight distributions for different scope experts.

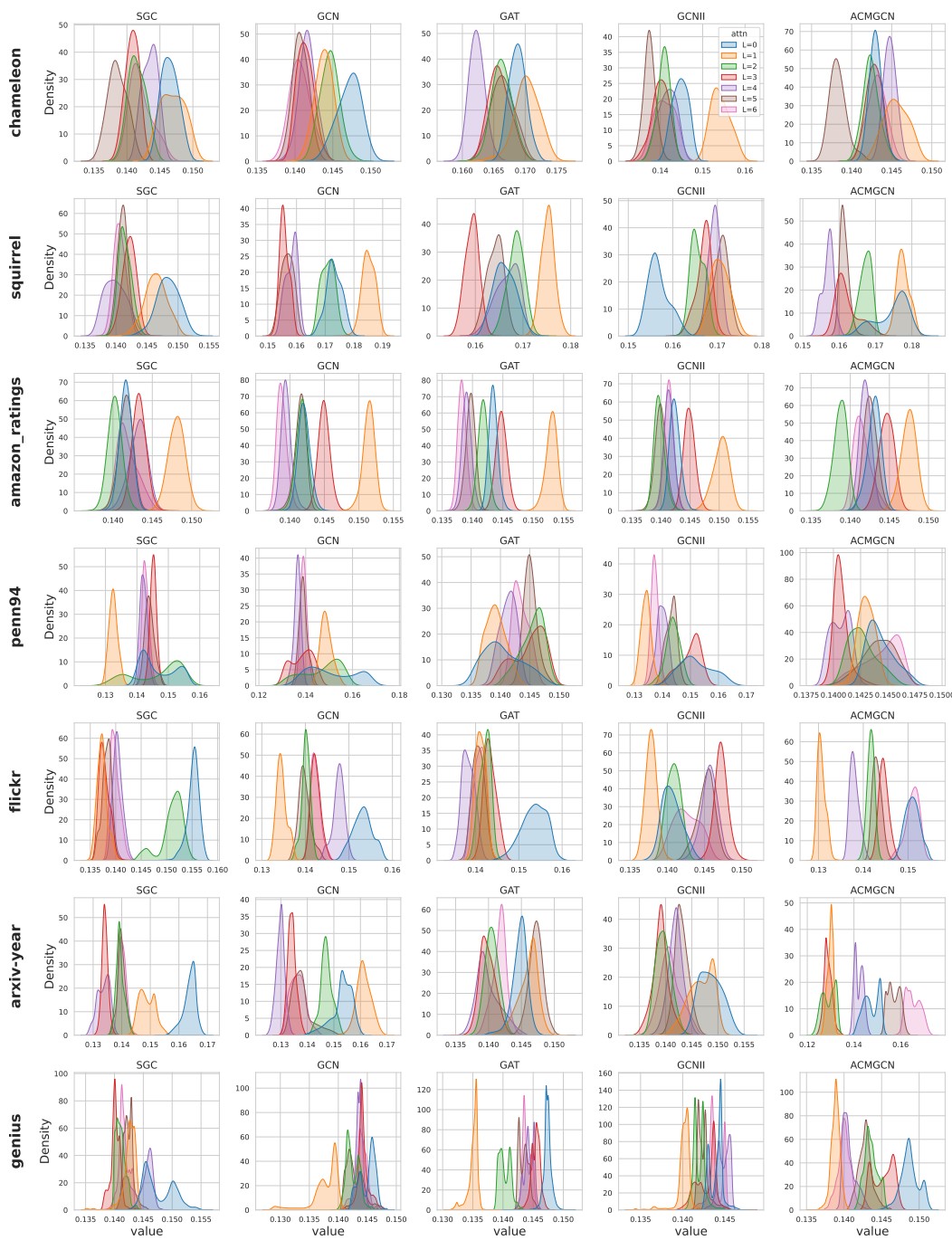

Figure 19: Learned attention-based gating weight distributions on heterophilous datasets. Different color denotes the weight distributions for different scope experts.

learns more *personalized* weights for shallow experts, while it adopts a more *uniform* weighting scheme for deeper experts. One plausible explanation for this trend is that increasing the scope size leads to a higher likelihood of overlapping neighbors among different nodes. Therefore, the gating model tends to learn the same weight on deeper experts for these nodes.

