# OpenReview forum: "Mixture of Scope Experts at Test: Generalizing Deeper Graph Neural Networks with Shallow Variants"
_NeurIPS.cc/2025/Conference — NeurIPS 2025 poster_

### Official Review · Reviewer_Hr9P · 2025-06-28

**Clarity:** 3
**Significance:** 2
**Originality:** 3
**Rating:** 5
**Confidence:** 4

**Summary:**

This paper investigates the problem of deeper GNNs theoretically and empirically, from which tries to propose a post-processing method of mixture of experts to improve the generalization capability while maintaining the high expressiveness. The proposed method is verified with extensive experiments that have been reported in Appendix.

**Questions:**

- How can you decide the optimal number of experts automatically?
- Is the implementation of comparing methods fully conducted to the original proposals?
- What is the limitation of the proposed method? It should be discussed at the conclusion for the future work.

**Ethical Concerns:**

["NO or VERY MINOR ethics concerns only"]

**Final Justification:**

After reading all the rebuttals, I'm satisfied with them, and retain my original score.

**Limitations:**

The same as above.

**Quality:**

4

**Strengths And Weaknesses:**

* Strengths
- Sound baseline of theoretical investigation on the problem of deeper GNNs.
- Reasonable proposal of the proposed method.
- Extensive evaluation with large variations of datasets and metrics.

* Weaknesses
- Optimal number of experts and deep and shallow models should be presented in detail.
- The main text needs more refined presentation. Especially, the figures need to be replaced with readable size and format.

---

> ### Author Rebuttal · Authors · 2025-07-31
>
> We are grateful for the reviewer’s thoughtful remarks and support for our work. Our detailed responses are as follows.
>
> ## Weaknesses and Questions
>
> > Optimal number of experts and deep and shallow models should be presented in detail.
> How can you decide the optimal number of experts automatically?
> >
>
> We use a hyperparameter $L_{\text{max}}$ to determine the maximum depth of all experts. There are a total of $L_{\text{max}}+1$ experts with scope sizes ranging from 0-hop (MLP) to $L_{\text{max}}$-hop.
>
> In our main experiments, we tune $L_{\text{max}}$ from 1 to 6 and find that $L_{\text{max}} = 6$ achieves the best results in most settings (Table 10). We also observe that increasing $L_{\text{max}}$ beyond 6 can further improve accuracy (Appendix G.3). The optimal number of experts follows a clear pattern: **higher values of $L_{\text{max}}$ consistently yield better accuracy but at increased computational cost** (Figures 6, 7, 8). As we add more and deeper experts, the less accuracy improvement we can expect from each additional one. Our experiments show that $L_{\text{max}}=6$ provides a good balance between accuracy and computational overhead across all tested settings.
>
> > Is the implementation of comparing methods fully conducted to the original proposals?
> >
>
> Yes, for baseline implementations and hyperparameters, we strictly followed the original papers (either those proposing the baselines or those evaluating existing approaches on newly proposed datasets). We conducted a hyperparameter search using Optuna when specific hyperparameters weren't provided in the original work.
>
> Since Moscat is designed as a post-processing approach, we use exactly the same baseline models as experts in our main experiments. In Appendix G.6, we also investigate how intentionally altering expert training affects Moscat's performance.
>
> > The main text needs more refined presentation. Especially, the figures need to be replaced with readable size and format.
> >
>
> Thank you for your feedback. We will refine the paper's presentation based on all reviewers' comments and carefully adjust the figure sizes and formats to ensure better readability.
>
> ## Limitations
>
> We will include a dedicated section discussing the limitations of Moscat in our revised version. The main limitations are as follows:
>
> - **Increased resource cost:** Moscat introduces extra memory and computational overhead due to training and inference with multiple experts in parallel.
> - **Limited effectiveness in homophilous graphs:** The method is less effective on homophilous datasets compared to heterophilous ones. Please refer to our response to "Reviewer 2PbC W4" for details.
> - **Lack of expert variations beyond depth:** Our work does not explore how factors beyond depth impact mixing performance, such as alternative expert training strategies or model architectures.
> - **Heuristic design and hyperparameter sensitivity:** Moscat currently relies on several heuristic choices and requires additional hyperparameter tuning. Future work could focus on developing more principled, neural-based approaches for training set selection and architecture search (for both experts and the gating model) to improve robustness and automation.

---

> > ### Comment · Reviewer_Hr9P · 2025-08-08
> > **Thanks for the response**
> >
> > I appreciate your detailed response clarifying all my questions. I am satisfied with it, and retain my score to accept it.

---

> > > ### Author Response · Authors · 2025-08-08
> > >
> > > Thank you so much!

---

### Official Review · Reviewer_XLzJ · 2025-06-30

**Clarity:** 3
**Significance:** 3
**Originality:** 3
**Rating:** 5
**Confidence:** 4

**Summary:**

This paper investigates a key phenomenon in Graph Neural Networks (GNNs): with increasing depth, they successfully capture homophily from multi-hop neighborhoods, yet exhibit a generalization bias across nodes with different degrees of homophily. Building on this finding, we propose a novel GNN architecture inspired by the Mixture-of-Experts (MoE) framework. Extensive experimental results validate the efficacy of our proposed method.

**Questions:**

With many mathematical symbols and formulas, the paper ultimately concludes that some parts of the network should be shallower while others should be deeper. However, the transition from the theoretical discussion to the Mixture-of-Experts framework is not sufficiently motivated. From my perspective, it currently feels somewhat like the we wanted a theory and then developed one to fit. While this new perspective—linking network depth and heterophily—is interesting and novel, I would like to see a stronger connection established between homophily and network depth.

**Ethical Concerns:**

["NO or VERY MINOR ethics concerns only"]

**Final Justification:**

The idea of linking homophily and network depths is interesting. But based on both the current version of the paper and the authors’ rebuttal, I believe that the correlation (or namely, why would we put these things together to formulate an analysis) between receptive field and heterophily remains insufficiently substantiated.

**Limitations:**

Yes

**Quality:**

3

**Strengths And Weaknesses:**

Strengths

- The paper's innovative use of a Mixture-of-Experts (MoE) is interesting. This allows the GNN to learn adaptive depths for distinct "subgroups" of nodes (e.g., those with varying homophily levels), which is a clever solution to handle non-uniform graph structures.
- The proposed method exhibits superior flexibility and performance. These claims are well-supported by extensive and rigorous experiments that validate the architecture's effectiveness across various benchmarks.

Weakness

- For the motivation of the paper, particularly the first three paragraphs,
    - Figure 1 shows how different levels of homophily affect the performance of GNNs. When we intentionally change the graph structure to make it more or less homophilous, most standard GNNs respond in similar ways because their message-passing relies on homophily. However, I don’t think this experiment here motivates the proposed method well.
    - The authors claim that deeper GNNs can capture homophily over a larger scope, or in common terms, a larger receptive field. However, it is not clear why a larger scope would help GNNs on heterophilous graphs. This point is not sufficiently discussed in the paper, nor are any related works cited. Heterophily is a well-studied area with many existing approaches, such as heterophily-oriented GNNs and Graph Transformers. Most of these methods do not treat the receptive field as a primary tool for addressing heterophily.

---

> ### Author Rebuttal · Authors · 2025-07-31
>
> Thank you for your thoughtful feedback. We genuinely appreciate your encouragement and insights. Our detailed responses are as follows.
>
> ## Weaknesses
>
> > Figure 1 shows how different levels of homophily affect the performance of GNNs. When we intentionally change the graph structure to make it more or less homophilous, most standard GNNs respond in similar ways because their message-passing relies on homophily. However, I don’t think this experiment here motivates the proposed method well.
> >
>
> We want to clarify that Figure 1 presents GNN performance on real-world data **without any manual intervention**. Unlike many existing works (e.g., [1, 2, 3]) that intentionally alter graph structure to synthetically generate datasets with different **graph-level** homophily ratios, we calculate **node-level** homophily ratios using the original graph structure and then group nodes into different subgroups based on their homophily scores. This approach gives Figure 1 practical significance and provides strong motivation for our work.
>
> > The authors claim that deeper GNNs can capture homophily over a larger scope, or in common terms, a larger receptive field. However, it is not clear why a larger scope would help GNNs on heterophilous graphs. This point is not sufficiently discussed in the paper, nor are any related works cited. Heterophily is a well-studied area with many existing approaches, such as heterophily-oriented GNNs and Graph Transformers. Most of these methods do not treat the receptive field as a primary tool for addressing heterophily.
> >
>
> Thank you for your careful review. We will add more related citations to support this claim in our revised version.
>
> We respectfully argue that many existing works have already shown that a larger scope (usually beyond 2 or 3 hops) is essential for learning heterophilous graphs. Many of these methods build upon using a large scope and design various techniques to address the challenges that arise when the scope is large. For example, GPR-GNN [3] sets a learnable weight for each hop and uses a large scope size K=10 in all its experiments. GCNII [4] and G$^2$-GNN [5] are specifically designed for deep GNN architectures. MixHop [6] proposes adding higher-order graph convolution operators in each layer, which directly enlarges the receptive field. Additionally, both our work (Table 12) and existing research [7, 8] have demonstrated that heterophilous graphs generally prefer deeper GNNs.
>
> H$_2$GCN [2] also theoretically justifies that higher-order neighborhoods will be more homophilous on heterophilous graphs, which encourages GNNs to use a larger scope.
>
> ## Questions
>
> > With many mathematical symbols and formulas, the paper ultimately concludes that some parts of the network should be shallower while others should be deeper. However, the transition from the theoretical discussion to the Mixture-of-Experts framework is not sufficiently motivated. From my perspective, it currently feels somewhat like the we wanted a theory and then developed one to fit. While this new perspective—linking network depth and heterophily—is interesting and novel, I would like to see a stronger connection established between homophily and network depth.
> >
>
> Moscat stems directly from our theoretical findings. In our revised paper, we will clarify the narrative that connects these theoretical insights to the specific architectural choices we made in our design. In the following, we will explain the logic flow of connecting homophily and network depth in the paper. Below, we outline the logical progression that links homophily and network depth in our paper.
>
> 1. In Section 1, we recall that a larger scope is required for addressing heterophily. We will add more citations in our refined version to justify this claim.
> 2. In Section 3.1, we dissect the challenges that GNNs face when tackling a larger scope. We reveal that the trained GNN model should be both expressive enough to capture higher-order patterns and generalizable enough to avoid overfitting. Although existing approaches for deeper GNNs can ensure expressivity and achieve close to 100% training accuracy, they suffer from poor generalization and may even fail to achieve better test accuracy than their shallow variants.
> 3. To investigate this phenomenon, Section 3.2 presents a theoretical analysis and concludes that (1) the test nodes can be categorized into low and high homophily regions based on the training set's average homophily ratio. (2) As network depth/scope increases, the model's generalization capability exhibits distinct trends in these two regions (indicating a shift in generalization pattern). (3) This insight explains why simply increasing GNN depth often fails to improve accuracy.
> 4. The above analysis directly motivates our method design:
>     1. From a higher level, this identified "generalization pattern shift across depths" (Section 3.2) motivates and supports our train-then-merge MoE framework for GNN models with various depths. In other words, rather than investigating how to train GNNs from scratch, we focus on learning the generalization patterns of already trained GNN models in a post-training stage.
>     2. From a lower level, the specific challenges that deeper GNNs encounter (Section 3.1) motivate our design choices:
>         - **Over-smoothing:** Our "Structural Encoding" technique enables the gating model to assess the likelihood that an expert for a given scope will experience over-smoothing.
>         - **Model degradation (Underfitting):** As discussed in the "Heterophily-Biased Sample Filtering" paragraph, we select samples in experts' training set $\mathcal{V}\_{\text{exp}}$ to form the gating model training set $\mathcal{V}\_{\text{gate}}$, which enables the gating model to learn how well each expert fits nodes with various structural patterns.
>         - **Overfitting:** We also use the holdout set $\mathcal{V}\_{\text{hold}}$ to form $\mathcal{V}\_{\text{gate}}$, which enables the gating model to detect experts' overfitting. We further introduce “Label Embeddings” to approximate node homophily, which can predict experts' generalization patterns (Theorem 1).
>
>
> [1] Characterizing Graph Datasets for Node Classification: Homophily-Heterophily Dichotomy and Beyond
>
> [2] Beyond Homophily in Graph Neural Networks: Current Limitations and Effective Designs
>
> [3] Adaptive Universal Generalized PageRank Graph Neural Network
>
> [4] Simple and Deep Graph Convolutional Networks
>
> [5] Gradient Gating for Deep Multi-Rate Learning on Graphs
>
> [6] MixHop: Higher-Order Graph Convolutional Architectures via Sparsified Neighborhood Mixing
>
> [7] A critical look at the evaluation of GNNs under heterophily: Are we really making progress?
>
> [8] Classic GNNs are Strong Baselines

---

> > ### Comment · Reviewer_XLzJ · 2025-08-05
> >
> > I thank the authors for their clarification, and I urge them to include detailed settings for Figure 1 in the next version of the paper.
> > Regarding the correlation between receptive field and heterophily, I regret to say that I am not convinced by the reasons provided. While it is true that many heterophily-related works introduce higher-order message passing, or equivalently, larger receptive fields, this is not their primary contribution. Rather, these methods focus on how to effectively utilize information from nodes at various hops, not simply on increasing the receptive field. For example, stacking multiple layers of GCNs does indeed increase the receptive field, but this alone does not address the issues associated with heterophily. Furthermore, the discussion in Section 3.1 appears to be a general argument that could apply to any deep learning method, rather than specifically to the context of this work.
> >
> > Based on both the current version of the paper and the authors’ rebuttal, I believe that the correlation between receptive field and heterophily remains insufficiently substantiated.

---

> ### Author Response · Authors · 2025-08-07
>
> Thank you for your thoughtful feedback. We will clarify the setup of Figure 1 in the revised version to ensure there is no ambiguity.
>
> Indeed, our work bridges the connection between two critical fields in graph learning: **deeper GNNs** and **learning on heterophily**. On one side, while many heterophily-related works leverage larger receptive fields, they provide limited insight into **why** this benefits learning on heterophilous graphs. On the other side, existing research explains why deeper GNNs are expressive and why they often fail to deliver performance improvements, but these explanations rarely consider the heterophily learning perspective.
>
> Section 3.1 comes from the perspective of deeper GNNs, summarizing the well-known reasons for their failure (over-smoothing, model degradation, and overfitting) in a unified framework and highlighting their trade-offs. However, due to the extremely limited space in the main text, we defer more GNN-related evidence into Appendix D. We will address this organization issue in our revised version.
>
> To understand **why and how larger scope benefits heterophily learning**, Section 3.1 and Appendix D draw from existing work to conclude that **deeper GNNs can ensure better expressivity but may suffer from poor generalization**. Building on this foundation, Section 3.2 presents new theoretical analysis examining how deeper GNNs generalize across different homophily regions, ultimately showing that **deeper GNNs can generalize better than shallow GNNs on node subgroups with specific homophily ratio ranges**. We will explicitly highlight this contribution in our paper and strengthen the narrative flow and connections between sections in our revised version.

---

### Official Review · Reviewer_rzM3 · 2025-06-30

**Clarity:** 3
**Significance:** 2
**Originality:** 2
**Rating:** 3
**Confidence:** 4

**Summary:**

This paper studies the generalization of deep layer GNNs and find deeper GNNs achieve better generalization than their shallow variants in either homophily or heterophily region, they sacrifice generalization performance in the other region. Then, the authors propose a mixture of experts model to select GNNs with different layers for different nodes. Experimental results demonstrate the proposed method can be applied in to different backbones.

**Questions:**

Please refer to the weaknesses.

**Ethical Concerns:**

["NO or VERY MINOR ethics concerns only"]

**Final Justification:**

I have thoroughly read the authors’ rebuttal. While they have addressed some of my concerns, I still question the novelty of their work. Moreover, based on my own experiments, I remain concerned about the effectiveness of their homophily estimation. I have updated my rating to borderline reject.

**Limitations:**

The authors should illustrate how do they select the heterophilous nodes and the performance on the low labeling rate.

**Quality:**

2

**Strengths And Weaknesses:**

Strengths:
1. The problem of generalization disparity of deep GNNs is worth to explore.
2. The paper is well-written and easy to follow.

Weaknesses:
1. The novelty is limited. The generalization disparity of deep gnns has been studied in several papers, for example, GMoE [1] also argue that different nodes require different hop of information. The theoretical analysis majorly follows [2], and I did not find something new here. Also, the MoE in GNNs is also utilized in several papers. For example, Link-MoE [3] also first train different GNN experts and then train the gating model to select different experts for different samples.

2. The authors filter out heterophilous nodes when training the experts. However, the heterophily of nodes are unknown due to the lack of ground-truth. How do the authors select the heterophilous nodes?

3. The authors leverage a high labeling rate in the experiments, which is impractical in the real-world applications. Can the proposed method work well when the labeling rate is low?

[1] Graph Mixture of Experts: Learning on Large-Scale Graphs with Explicit Diversity Modeling

[2] Demystifying structural disparity in graph neural networks: Can one size fit all?

[3] Mixture of Link Predictors on Graphs

---

> ### Author Rebuttal · Authors · 2025-07-31
>
> We thank the reviewer for the time and effort spent reviewing our paper. We appreciate the opportunity to address any concerns and clarify potential misunderstandings. Our detailed responses are as follows.
>
> ## W1
>
> > The generalization disparity of deep gnns has been studied in several papers, for example, GMoE [1] also argue that different nodes require different hop of information.
> >
>
> Exploring generalization disparities of GNNs across varying depths is crucial and creates new opportunities to advance this field. However, we argue that **this question remains largely unexplored**.
>
> Existing studies like GMoE and ACMGCN primarily focus on designing expressive architectures (e.g., node-adaptive hop aggregation) to better fit training data rather than investigating generalization disparity. In contrast, our work specifically examines generalization, and to the best of our knowledge, we are the first to formally investigate "How GNNs of different **depths** generalize across **node subgroups**." We also establish novel and significant conclusions, such as "Varying the depth shifts the GNN generalization pattern across node subgroups."
>
> Our method's substantial performance improvements over these existing approaches provide strong empirical validation for the significance of our findings.
>
> > The theoretical analysis majorly follows [2], and I did not find something new here.
> >
>
> Paper [2] provides a valuable foundation regarding how GNNs generalize across nodes with different structural properties. However, while [2] focuses on the generalization disparities between **GNN vs MLP**, our work examines the differences between **shallow vs deeper GNN**. Although [2] establishes a mathematical framework to study GNNs' generalization gap, we extend this framework **beyond 1-layer GNNs** to investigate how depth variation affects GNN generalization performance on nodes with various structural properties. This extension is crucial because depth is the most critical parameter of GNNs, determining the model's receptive field and fundamentally distinguishing GNNs from other neural networks.
>
> > Also, the MoE in GNNs is also utilized in several papers. For example, Link-MoE [3] also first train different GNN experts and then train the gating model to select different experts for different samples.
> >
>
> Coincidentally, both Moscat and Link-MoE share the same insight that "train-then-merge" MoEs are more suitable for graph learning than expert-gate joint training MoEs (please refer to our Appendix E for more details). However, Link-MoE is based on empirical heuristics found in link prediction tasks (e.g., heuristics link predictors like common neighbors show different generalization patterns than GNN link predictors), while Moscat's design is to address the prevalent yet critical generalization issue of deeper GNNs in heterophilous graphs, backed by solid mathematical foundations. They have drastically different targets, assumptions, and model designs.
>
> Beyond sharing the same "train-then-merge" philosophy with Link-MoE, our work bridges two key areas in graph learning: (1) Deeper GNN and (2) Graph MoE. As shown in Figure 2, we connect "expert-gate joint training" MoE design to GNN "soft scoping" and "train-then-merge" MoE to GNN "hard scoping". We further justify why "hard scoping" design is more suitable for GNNs, a critical insight rarely discussed in previous literature.
>
> ## W2
>
> Recall that the definition of node homophily used in our paper is
>
> $$
> \mathbf{h}_v^{\text {node }}=\left|\left\\{u \in \mathcal{N}_v: y_u=y_v\right\\}\right| /\left|\mathcal{N}_v\right|,
> $$
>
> where $y_v$ is the label for node $v$ and $\mathcal{N}_v$ denotes $v$'s neighbors. We aim to calculate the homophily score $\mathrm{h}^{\text {node }}\_v$ for every node $v\in \mathcal{V}\_{\text{train}}$ in the training set. However, the label for neighbor node $u\in\mathcal{N}_v$ is not available when $u$ is in the test set $u\in \mathcal{V\_{\text{test}}}$. To address this issue, we remove all edges connected to unlabeled nodes (i.e., remove all the test nodes from $\mathcal{N}_v$) before calculating the homophily ratios for training nodes. Hence, there is strictly no information leakage, and the comparison is fair.
>
> Please refer to run_moscat.py (lines 411-422) in our supplementary material for detailed implementation. In our revised version, we add a note to clarify this approximation of the homophily score.
>
> ## W3
>
> Thank you for this insightful comment. Since most existing studies (e.g., [1, 3, 4]) on heterophilous graphs employ high labeling rates, we adopted this standard setting to directly compare with prior work in our paper. We have now added experiments for low labeling rate scenarios below.
>
> For these additional experiments, we maintain the same test sets from the original paper and vary the proportion of available labels for training and validation from 20% to 100%, with 100% representing the setup used in our paper. All other experimental parameters remained unchanged.
>
> |  | Labeling Ratio | 20% | 40% | 60% | 80% | 100% |
> | --- | --- | --- | --- | --- | --- | --- |
> | Amazon-Ratings | GCN | 41.25 ±1.08 | 44.44 ±0.56 | 45.98 ±0.59 | 47.39 ±0.53 | 48.55 ±0.38 |
> |  | GCN-Moscat | 43.42 ±0.75 | 46.50 ±0.73 | 48.86 ±0.37 | 50.54 ±0.64 | 52.25 ±0.69 |
> |  |  |  |  |  |  |  |
> | Penn94 | GCN | 74.00 ±0.69 | 77.65 ±0.52 | 80.11 ±0.43 | 81.35 ±0.53 | 82.54 ±0.43 |
> |  | GCN-Moscat | 78.17 ±1.05 | 81.95 ±0.81 | 83.71 ±0.54 | 84.75 ±0.59 | 85.76 ±0.32 |
> |  |  |  |  |  |  |  |
> | Flickr | GCN | 52.49 ±0.34 | 53.60 ±0.42 | 54.51 ±0.28 | 54.93 ±0.25 | 55.52 ±0.36 |
> |  | GCN-Moscat | 54.49 ±0.49 | 55.52 ±0.34 | 56.61 ±0.30 | 57.46 ±0.17 | 57.75 ±0.16 |
>
> Our results show that Moscat consistently outperforms a single expert across different labeling rates by a significant margin. As training sample sizes vary, experts may suffer from overfitting or underfitting. Moscat's decoupled gating design effectively handles both scenarios.
>
> The results also reveal three distinct patterns: (1) For Amazon-Ratings, the performance gap between GCN-Moscat and GCN widens as the labeling ratio increases. (2) For Penn94, this gap narrows with higher labeling ratios. (3) For Flickr, the performance difference remains consistent across all labeling ratios.
>
> To understand these patterns, we follow the methodology in Appendix G.6 to measure the percentage of nodes that are misclassified (all_wrong) or correctly classified (all_correct) by all GCN experts:
>
> |  | Labeling Ratio | 20% | 40% | 60% | 80% | 100% |
> | --- | --- | --- | --- | --- | --- | --- |
> | Amazon-Ratings | all_wrong | 28.01 | 27.49 | 26.07 | 25.24 | 24.62 |
> |  | all_correct | 9.12 | 10.96 | 11.33 | 11.84 | 12.22 |
> |  |  |  |  |  |  |  |
> | Penn94 | all_wrong | 5.11 | 4.06 | 3.38 | 3.10 | 2.88 |
> |  | all_correct | 29.07 | 33.11 | 34.67 | 36.08 | 36.61 |
> |  |  |  |  |  |  |  |
> | Flickr | all_wrong | 31.26 | 30.86 | 29.74 | 29.51 | 29.57 |
> |  | all_correct | 25.48 | 26.83 | 26.54 | 27.11 | 27.37 |
>
> Based on the above table, we observe:
> 1. On Amazon-Ratings, "all_wrong" significantly decreases as the labeling ratio increases. This suggests that experts can better specialize in their respective domains with more labeled data, which enhances performance after mixing.
> 2. On Penn94, "all_wrong" slightly decreases while "all_correct" increases significantly as the labeling ratio increases, indicating that experts become more accurate and overlapped when trained with more labeled data. This increased overlap reduces Moscat's effectiveness.
> 3. On Flickr, both "all_wrong" and "all_correct" remain relatively consistent across labeling ratios, which explains why Moscat's improvements are also relatively stable.
>
>
> [1] Large Scale Learning on Non-Homophilous Graphs: New Benchmarks and Strong Simple Methods
>
> [2] Demystifying structural disparity in graph neural networks: Can one size fit all?
>
> [3] A critical look at the evaluation of GNNs under heterophily: Are we really making progress?
>
> [4] Revisiting Heterophily For Graph Neural Networks

---

> > ### Author Response · Authors · 2025-08-02
> >
> > Dear Reviewer rzM3,
> >
> > Thank you for your time and thoughtful feedback on our paper. We hope our response has addressed your concerns. If there are any remaining questions, we would be grateful for the opportunity to provide further clarification.

---

> > ### Comment · Reviewer_rzM3 · 2025-08-05
> >
> > Thanks for the authors response, which have addressed most of my concerns. I still have concern about the node homophily calculation. You said the node homophily is calculated only by the training nodes. However, If all val/test nodes are removed, the graph might be disconnected, especially when the labeling rate is low. How to calculate the node homophily for isolated node? Besides, can the proposed method work on the semi-supervised setting for homophilic graphs?

---

> > > ### Author Response · Authors · 2025-08-05
> > >
> > > Thank you for your response and follow-up questions. We are pleased to have the opportunity to provide further clarification.
> > >
> > > **Homophily Estimation**
> > >
> > > When calculating the estimated homophily score, we only consider the 1-hop neighborhood. Therefore, graph disconnection does not affect our estimation. Let $\mathcal{N}_v$ denote $v$'s full neighbor node set, and $\mathcal{N}^{\text{train}}_v$ denote the labeled nodes in $v$'s neighborhood. $\mathcal{N}^{\text{train}}_v$ can be considered as randomly sampled from $\mathcal{N}_v$, and we use it as a Monte Carlo approach to approximate the true homophily score of node $v$. As long as $|\mathcal{N}^{\text{train}}_v|$ is sufficiently large, we can obtain a reliable estimation of $v$'s homophily score without requiring a high labeling ratio (i.e., $|\mathcal{N}^{\text{train}}_v|/ |\mathcal{N}_v|$). Additionally, we only use the estimated homophily score for binary classification to determine whether a node is homophilous or heterophilous, which doesn't require extreme precision.
> > >
> > > For isolated nodes or nodes with small $|\mathcal{N}^{\text{train}}_v|$, we cannot obtain an accurate estimation of homophily score. However, these estimation errors are tolerable in practice. Below, we quantify our estimation's accuracy by calculating the percentage of heterophilous nodes correctly identified:
> > >
> > > |  | Amazon-Ratings | Ogbn-Arxiv |
> > > | --- | --- | --- |
> > > | Percentage of oracle heterophilous nodes correctly identified by the estimated homophily score | 91.43% | 95.66% |
> > > |  |  |  |
> > >
> > > These results demonstrate that our estimation can correctly identify most heterophilous nodes. We further validate our approach by comparing task performance:
> > >
> > > |  | Amazon-Ratings | Ogbn-Arxiv |
> > > | --- | --- | --- |
> > > | GCN-Moscat w/ estimated homophily (Ours) | 52.25 ±0.69 | 74.09 ±0.32 |
> > > | GCN-Moscat w/ oracle homophily | 53.36 ±0.65 | 74.19 ±0.25 |
> > > |  |  |  |
> > >
> > > On the Ogbn-Arxiv dataset, where homophily estimation is more accurate (95.66% vs. Amazon-Ratings' 91.43%), our method achieves accuracy (74.09 ±0.32) nearly identical to that using oracle homophily (74.19 ±0.25). We observe a larger performance gap in the Amazon-Ratings dataset due to its less accurate homophily estimation. In special scenarios with extremely low label availability, we may not observe accuracy improvements from filtering estimated heterophilous nodes. However, our method still maintains strong performance even without applying this technique.
> > >
> > > **Moscat’s Performance on Homophilous Graphs**
> > >
> > > We have added further experiments to evaluate the performance of Moscat on homophilous graphs in our response to "Reviewer 2PbC W4". We paste it below for your convenience:
> > >
> > > Our paper already includes results from two homophilous datasets: Ogbn-Arxiv and Pubmed. To address your suggestion, we have evaluated Moscat on two additional widely used homophilous datasets, WikiCS and Amazon Computers, presented below:
> > >
> > > |  | Ogbn-Arxiv | WikiCS | Pubmed | Computers | Avg. Improv. |
> > > | --- | --- | --- | --- | --- | --- |
> > > | Avg. Homo. Ratio | 0.64 | 0.66 | 0.79 | 0.80 |  |
> > > | GCN | 72.13 ±0.17 | 80.71 ±0.52 | 90.15 ±0.71 | 91.76 ±0.39 |  |
> > > | +Global Ensemble | 73.16 ±0.11 | 81.18 ±0.46 | 91.37 ±0.62 | 92.00 ±0.36 | $\Uparrow$0.91% |
> > > | +Moscat | 74.09 ±0.32 | 82.39 ±0.46 | 91.97 ±0.44 | 94.27 ±0.49 | $\Uparrow$2.39% |
> > >
> > > We follow the same settings as in our paper and introduce a new baseline "Global Ensemble," which uses Bayesian optimization to tune the global ensemble weights on the validation set. The results show that Moscat also works well on homophilous datasets, achieving a $\Uparrow$2.39% average$%$ improvement, which is significantly more effective than Global Ensemble with only $\Uparrow$0.91% improvement. However, there's still a notable gap compared to the $\Uparrow$5.28% average improvement that Moscat achieves on the heterophilous datasets tested in our paper.
> > >
> > > Additionally, there exist datasets with even higher homophily ratios (e.g., Amazon Photo at 0.85 and Coauthor-CS at 0.92). However, a single GNN can already achieve over 95% accuracy on these datasets. These datasets are likely too simple and therefore inappropriate for evaluating our method.

---

> > > > ### Author Response · Authors · 2025-08-08
> > > >
> > > > Thank you again for your thoughtful follow-up questions.
> > > > I’ve added further clarification and experiments regarding the homophily estimation and Moscat’s performance on homophilous graphs in my latest response above.
> > > > As the discussion period is wrapping up, I’d be happy to provide any additional details if that would be helpful.

---

### Official Review · Reviewer_2PbC · 2025-07-06

**Clarity:** 3
**Significance:** 4
**Originality:** 3
**Rating:** 5
**Confidence:** 4

**Summary:**

The authors propose Moscat, a framework for training a mixture of GNNs that addresses existing limitations of over-smoothing and generalization. Different from previous work relying on mixtures of experts (MoE) where experts are trained jointly together with an agreggation module, Moscat proposes to train experts separately (and possibly simultaneously), and then train a gating module aimed at combining their strengths and improving generalization. The authors present extensive theoretical and empirical findings motivating Moscat, as well as experimental results that demonstrate the effectiveness of their proposed approach.

**Questions:**

1. How much more expensive is it to run Moscat in terms of memory usage, in comparison with a single GNN?
2. Is Moscat robust to the way the data is split for generating the hold-out set used to train the gating model?
3. When faced with new datasets in real world problems, how would you advice practitioners starting with Moscat to make the large number of options (underlying GNN architecture, number of layers, hold-out set, heterophily-biased sampling, scope-aware logit augmentation, plus tunable hyperparameters) more manageable?
4. Could you please clarify which parts of Moscat adress each of the challenges listed at the beginning of Section 4 (L217-226)?

**Ethical Concerns:**

["NO or VERY MINOR ethics concerns only"]

**Final Justification:**

The authors have addressed all my remarks and questions during the rebuttal and discussion periods. These even led to additional insights that I hope the authors can incorporate in their work. I therefore keep my recommendation for acceptance.
I think this is a solid contribution, and as such I would be willing to champion this paper if necessary.

**Limitations:**

The authors do not discuss the limitations of Moscat (apart from a short sentence in the Checklist), which justifies my Quality score. I would advice the authors to reflect on limitations such as cost, and the lack of a more principled approach, which could help understand the method as well as making future directions of research more clear.

**Paper Formatting Concerns:**

No concerns.

**Quality:**

3

**Strengths And Weaknesses:**

**Strengths**

1. The authors provide a comprehensive motivation for their approach, rooted in issues like over-smoothing, vanishing gradients, and overfitting. Theorem 3 is insightful as it points to an explanation for the low performance of GNNs in some cases, involving disparity of generalization between homophilous and heterophilous groups of nodes at different layers of a GNN.
2. The previous result is accompanied by experiments that support the low overlap between correctly classified nodes across different layers of a GNN.
3. In spite of several heuristics included in Moscat, the overall approach is simple and intuitive: instead of designing specific GNNs, or jointly training experts, train separately L experts and then train a gating model. This is useful because the framework can be applied to a broad class of GNNs, as evidenced in the experiments.
4. The approach incorporates heuristics that are well motivated and effective, such as heterophily-biased sample filtering and scope-aware logit augmentation.
5. The improvements brough by Moscat are clear and significant, both when examining the improvements over the same architecture, and in comparison with strong baselines (including MoE approaches).
6. The authors provide ablation experiments that justify the different parts of Moscat (Table 4) and interesting interpretation experiments in the appendix.

**Weaknesses**

1. The fundamental message behind Moscat is that in general, a single deep GNN is almost guaranteed to perform poorly on node classification. Therefore, the solution is to train several GNNs of different depths, while learning how to combine them properly. This means that the cost of training Moscat is always larger than the cost of a single GNN.  Runtime analysis is deferred to the appendix, though I believe it should play a more central role; but more importantly it should also include a **space complexity** analysis, together with experimental results, that answers the question: how much more expensive is it to run Moscat? This would help contextualize if the observed gains are worth the cost.
2. The hold-out set seems important for training the gating model, but it seems that the authors adopt a single static way to generate this split. Whether this is a source of variation in performance is not clear.
3. The approach is composed by several moving parts (underlying GNN architecture, number of layers, hold-out set, heterophily-biased sampling, scope-aware logit augmentation, plus tunable hyperparameters), rather than a more principled solution to the issues of GNNs discussed and found by the authors. I would expect that this would limit the practical impact of Moscat due to increased systems complexity and optimization cost, in contrast with a more principled solution.
4. Even though the paper is focused on the case of heterophilous graphs, it would have been interesting to see if Moscat also works well on homophilous graphs. The closest case is Pubmed, but extending these experiments would further increase our understanding of how performant and broadly applicable Moscat is.

---

> ### Author Rebuttal · Authors · 2025-07-31
>
> We sincerely thank the reviewer for the encouraging and constructive feedback.
> Our detailed responses are as follows.
>
> ## W1 / Q1
> Thank you for your constructive comments. We agree that space consumption should be carefully considered.
>
> Moscat allows each expert and the gating model to be trained separately. Practitioners can flexibly adjust the parallelism of expert training to balance runtime and memory consumption. Let $L_\text{max}$ denote the maximum depth of GNN experts, $F_{\text{hid}}$ denote the hidden dimension, and $\mathcal{V}\_{\mathrm{exp}}$  and  $\mathcal{V}\_{\mathrm{gate}}$ denote the training sets for experts and the gating model, respectively. Taking GCN as an example, we have two boundary cases:
>
> - If we prioritize minimal memory overhead, we can train each expert sequentially. The space complexity is therefore bounded by training the deepest GNN expert: $O\left(L\_{\text{max}}|\mathcal{V\_{\text{exp}}}|F\_{\text{hid}} + L\_{\text{max}}F\_{\text{hid}}^2\right)$.
> - If we prioritize minimal runtime, we can train all experts in parallel. The space complexity then becomes proportional to the total number of layers ($1 + 2 + \cdots + L\_{\text{max}}$) across all experts: $O\left(L\_{\text{max}}^2|\mathcal{V}\_{\text{exp}}|F_{\text{hid}} + L\_{\text{max}}^2F\_{\text{hid}}^2\right)$
>
> The memory consumption of the gating model training is relatively small and won't become a bottleneck: $O\left(L\_{\text{max}}|\mathcal{V\_{\text{gate}}}|F\_{\text{hid}} + L\_{\text{max}}F\_{\text{hid}}^2\right)$, since $F\_{\text{hid}} \ll|\mathcal{V}\_{\text{gate}}|\ll|\mathcal{V}\_{\text{exp}}|$.
>
> **In conclusion, with comparable runtime, Moscat requires approximately $L\_{\text{max}}$ times more memory than a single $L\_{\text{max}}$-layer GNN during training.**
>
> We also measured the peak GPU memory consumption for training GCN and GCN-Moscat with $L\_{\text{max}}=6$ as shown in the following table:
>
> |  | L=0 (MLP) | L=1 | L=2 | L=3 | L=4 | L=5 | L=6 | Total (Moscat) |
> | --- | --- | --- | --- | --- | --- | --- | --- | --- |
> | amazon-ratings | 1294MB | 618MB | 980MB | 1194MB | 1418MB | 1616MB | 1832MB | 8952MB |
> | arxiv-year | 1164MB | 772MB | 1026MB | 1240MB | 1474MB | 1688MB | 1922MB | 9286MB |
>
> On amazon-ratings, GCN-Moscat ($L\_{\text{max}}=6$) requires 8952MB, which is approximately **4.9x** more than GCN (L=6) at 1832MB. The arxiv-year dataset also shows a similar ratio.
>
> ## W2 / Q2
>
> In the paper, we examine Moscat on two different setups for the holdout set (Section 5.1):
>
> (1) For "Moscat", we use 10% of the original validation set for validating the gating model and the rest for gating model training. We found the accuracy is stable by tuning this ratio between 0%~30%. The accuracy drops gradually by further increasing the ratio beyond 30%. Hence, we fix it to be 10% for all datasets.
>
> (2) For "Moscat$\ast$", we don't use the original validation set for expert/gate training. Instead, we partition a holdout set from the original training set. We introduce a hyperparameter $\alpha$ to represent the proportion of nodes for expert training, and the remaining $1-\alpha$ portion serves as the holdout set. Experts have fewer training samples and achieve lower accuracy with a lower $\alpha$, but the gating model can better capture experts' generalization patterns. Experts achieve higher accuracy with a higher $\alpha$, but the gating model may be less effective. In practice, we found that the downstream accuracy achieves the best and is relatively stable when $\alpha$ is between 0.55 $\sim$ 0.9. The table below shows the accuracy of GCN experts and GCN-Moscat$\ast$ with different $\alpha$ values on Penn94. We observed that while expert accuracy decreases as $\alpha$ decreases, the overall accuracy remains relatively stable.
>
> | $\alpha$ | 0.6 | 0.65 | 0.70 | 0.75 | 0.80 | 0.85 | 0.90 |
> | --- | --- | --- | --- | --- | --- | --- | --- |
> | GCN | 80.03 ±0.37 | 80.66 ±0.33 | 80.90 ±0.45 | 81.14 ±0.26 | 81.36 ±0.38 | 81.62 ±0.28 | **82.02** ±0.34 |
> | GCN-Moscat$\ast$ | 84.87 ±0.55 | 85.01 ±0.55 | 85.08 ±0.51 | 85.12 ±0.39 | 85.38 ±0.32 | **85.51 ±0.28** | 85.43 ±0.34 |
>
> We notice that Moscat$\ast$ often achieves the best accuracy with $\alpha$ around 0.85~0.90. Please refer to Appendix F.2 for more details.
>
> ## W3 / Q3
>
> The core reason for Moscat to have all these components is that we want it to be general to various datasets and GNN architectures. While Moscat has a large potential hyperparameter search space, we have established clear, principled best practices that significantly reduce this complexity. These guidelines, derived from our comprehensive experiments and analysis, make adapting Moscat to new datasets straightforward.
>
> 1. **Expressive GNN experts.** Although Moscat can adapt to various expert architectures, we observed that Moscat achieves better accuracy with more expressive GNN architectures (e.g., GAT, ACMGCN, MixHop), as shown in Table 1. This occurs because expressive architectures enable GNNs to maintain high training accuracy as depth increases, essential for learning patterns from higher-order neighbors. For details on how expert parameters affect Moscat's performance, please refer to Appendix G.6.
> 2. **Moscat hyperparameter tuning.** We organize the entire hyperparameters (hparams) space for the gating model training into three distinct categories:
>     1. **Hparams that do not need to be tuned.** By referring to Tables 6 and 8, we conclude that dropout=0, weight decay=0, number of MLP layers=3, and hidden_dim=256 work well for all settings. The maximum depth $L_{\text{max}}$ has also been extensively studied in Appendix G.3. The conclusion is straightforward: higher $L_{\text{max}}$ can achieve higher accuracy but incurs higher cost. We found $L_{\text{max}}$=6 achieves a good accuracy-overhead balance across all settings.
>     2. **Hparams that require minimal tuning for near-optimal performance.** Learning rate is crucial, as each dataset has its specific range of effective learning rates. Three other binary parameters worth tuning are: holdout-set-related ratio $\eta$, the "mask wrong" flag, and batch/layer normalization. Since these three are all **binary** parameters, the search space remains small.
>     3. **Hparams that can achieve slight improvements with further tuning.** Adjusting the holdout-set-related ratio $\alpha$ and heterophily-biased sampling related ratio $\gamma$ beyond their default values can slightly improve performance on some datasets.
>
> For more comprehensive information, please refer to Appendices F.2 and F.4. Components such as scope-aware logit augmentation can always be employed as they introduce virtually no adverse effects. We will provide this guidance on GitHub with a more detailed, self-contained version to assist practitioners in implementation.
>
> ## W4
>
> Thank you for your constructive feedback. Our paper already includes results from two homophilous datasets: Ogbn-Arxiv and Pubmed. To address your suggestion, we have evaluated Moscat on two additional widely used homophilous datasets, WikiCS and Amazon Computers, presented below:
>
> |  | Ogbn-Arxiv | WikiCS | Pubmed | Computers | Avg. Improv. |
> | --- | --- | --- | --- | --- | --- |
> | Node Homo. | 0.64 | 0.66 | 0.79 | 0.80 |  |
> | GCN | 72.13 ±0.17 | 80.71 ±0.52 | 90.15 ±0.71 | 91.76 ±0.39 |  |
> | +Global Ensemble | 73.16 ±0.11 | 81.18 ±0.46 | 91.37 ±0.62 | 92.00 ±0.36 | $\Uparrow$0.91% |
> | +Moscat | 74.09 ±0.32 | 82.39 ±0.46 | 91.97 ±0.44 | 94.27 ±0.49 | $\Uparrow$2.39% |
>
> We follow the same settings as in our paper and introduce a new baseline "Global Ensemble," which uses Bayesian optimization to tune the global ensemble weights on the validation set. The results show that Moscat also works well on homophilous datasets, achieving a $\Uparrow$2.39% average$%$ improvement, significantly more effective than Global Ensemble with only $\Uparrow$0.91% improvement. However, there's still a notable gap compared to the $\Uparrow$5.28% average improvement that Moscat achieves on the heterophilous datasets tested in our paper.
>
> Although there exist datasets with even higher homophily ratios (e.g., Amazon Photo at 0.85 and Coauthor-CS at 0.92), a single GNN can already achieve over 95% accuracy on these datasets. These datasets are likely too simple and therefore inappropriate for evaluating our method.
>
> ## Q4
>
> Section 4 (L217-226) lists the following three challenges:
>
> 1. "Feature construction" corresponds to "Scope-Aware Logit Augmentation".
> 2. "Training sample selection" corresponds to "Heterophily-Biased Sample Filtering".
> 3. We have dissected the adaptation to "Diverse Expert architecture" into the following three challenges:
>     1. **Over-smoothing:** As discussed in the "Scope-Aware Logit Augmentation" paragraph, our "Structural Encoding" technique enables the gating model to assess the likelihood that an expert for a given scope will experience over-smoothing.
>     2. **Model degradation (Underfitting):** As discussed in the "Heterophily-Biased Sample Filtering" paragraph, we select samples in experts' training set $\mathcal{V}\_{\text{exp}}$ to form the gating model training set $\mathcal{V}\_{\text{gate}}$, which enables the gating model to learn how well each expert fits nodes with various structural patterns.
>     3. **Overfitting:** We also use the holdout set $\mathcal{V}\_{\text{hold}}$ to form $\mathcal{V}\_{\text{gate}}$, which enables the gating model to detect experts' overfitting. We further introduce "Label Embeddings" to approximate node homophily, which can predict experts' generalization patterns.
>
> We will add clear pointers in our revised version to better structure Section 4.
>
> ## Limitations
>
> We appreciate the reviewer's suggestion and will include a dedicated section discussing Moscat's limitations. Due to the space limit, please refer to our response to "Reviewer Hr9P Limitations".

---

> > ### Comment · Reviewer_2PbC · 2025-08-04
> >
> > Thank you for the extensive response. You have sufficiently addressed most of my remarks and questions.
> > One that remains is with respect to the holdout validation set (W2/Q2). I understand that you investigated robustness by varying the percentage. But even for a fixed percentage, say 10%, there are many ways of splitting the labeled nodes and putting them in the holdout
> > This is the source of variance I was referring to. Is this variance indeed apparent when selecting the holdout validation set for Moscat?

---

> > > ### Author Response · Authors · 2025-08-05
> > >
> > > Dear Reviewer 2PbC,
> > >
> > > Thank you for your follow-up and further clarification.
> > >
> > > Since the holdout set is intended to evaluate the experts’ generalization performance, we expect it to follow the same distribution as the test set (i.e., the true distribution). To ensure this alignment, we use random sampling to select labeled nodes for the holdout set.
> > >
> > > In general, using alternative sampling strategies can shift the distribution of the holdout set away from the true distribution, which is not recommended and may lead to an apparent drop in accuracy. In such cases, the gating model may overfit to spurious patterns in the holdout set rather than learning to generalize effectively.
> > >
> > > While our design allows GNN experts of varying depths to overfit different structural patterns, the gating model is expected to calibrate the experts’ confidence and combine their predictions appropriately. This makes it especially important to construct the holdout set through random sampling to avoid distribution shift.

---

> > > > ### Comment · Reviewer_Hr9P · 2025-08-05
> > > > **Distribution of holdout set**
> > > >
> > > > I don't think that it's fair. If you choose the holdout set with the same distribution of the test set, it might be a sort of cheating to guide the model to fit to the test set.

---

> > > > > ### Author Response · Authors · 2025-08-05
> > > > >
> > > > > Dear Reviewer Hr9P,
> > > > >
> > > > > We hope our response has addressed your concern. Our holdout set is only randomly sampled from the labeled data, which does not depend on any test nodes. Therefore, it is totally fair without any data leakage.

---

> > > > > > ### Comment · Reviewer_2PbC · 2025-08-05
> > > > > >
> > > > > > I'm afraid my question is still not clear. Please allow me to rephrase it.
> > > > > > Let's consider random sampling as the strategy for selecting nodes for the holdout set (i.e. the strategy you used).
> > > > > > The selected nodes will change each time the (pseudo) random sampler is run (assuming that the seed changes at each time).
> > > > > > Then let's say that for i = 1, 2, ..., k
> > > > > > - We set the seed of the sampler to i
> > > > > > - We sample n nodes to form the holdout validation set $V_i$
> > > > > > - We train Moscat using $V_i$ as the holdout validation set and compute the resulting accuracy $a_i$
> > > > > >
> > > > > > The question is: what is the variance of the list of accuracies $(a_i)_{i=1}^k$? This could give us an idea of how robust Moscat is to the selected subset of nodes in the holdout set.

---

> > > > > > > ### Author Response · Authors · 2025-08-06
> > > > > > >
> > > > > > > Thank you for providing such a detailed example to clarify this point. Following the same setup as in our paper, we vary only the seed of the random sampler when selecting the holdout set:
> > > > > > >
> > > > > > > |  | seed=0 | seed=100 | seed=200 | seed=300 | seed=400 | seed=500 | Standard Deviation |
> > > > > > > | --- | --- | --- | --- | --- | --- | --- | --- |
> > > > > > > | Amazon-Ratings GCN-Moscat | 52.25 ±0.69 | 52.19 ±0.69 | 51.98 ±0.57 | 52.17 ±0.76 | 52.12 ±0.81 | 52.18 ±0.47 | 0.08 |
> > > > > > > | Penn94 GCN-Moscat | 85.76 ±0.32 | 85.68 ±0.58 | 85.62 ±0.49 | 85.62 ±0.54 | 85.59 ±0.39 | 85.75 ±0.44 | 0.07 |
> > > > > > >
> > > > > > > These results confirm that Moscat’s accuracy is robust to different holdout set selections, with only minimal variance across seeds. Notably, this variation is substantially smaller than that observed under different train/validation/test splits.

---

> > > > > > > > ### Comment · Reviewer_2PbC · 2025-08-06
> > > > > > > >
> > > > > > > > Thank you for your response. I'm glad to see new insights obtained after the discussion.
> > > > > > > > I have no further remarks or questions.

---

> ### Author Response · Authors · 2025-08-05
>
> I want to clarify that most commonly, we assume the train/test sets are randomly sampled from the true distribution, which means they share the same distribution. If we also randomly sample the holdout set from the existing labeled data (e.g., training set), we can say the holdout set follows the true distribution (i.e., the same distribution as the test set).
>
> Since we do not target the out-of-distribution datasets, we can say the holdout set shares the same distribution as the test set (and also the train set).

---

### Note · Authors · 2025-08-16

We sincerely thank all the reviewers for their thoughtful feedback and the considerable time and effort they devoted to reviewing our paper.

Our paper received 3 "Clear Accepts" and 1 "Reject," all with a high confidence score of 4. We are deeply grateful that the majority of the reviewers recognized the contributions of our work. We summarize the strengths of our work identified by the reviewers:
- Strong motivation and novelty for our proposed paradigm (Reviewers 2PbC, XLzJ, Hr9P).
- Valuable analysis on deeper GNNs' generalization disparities (Reviewers 2PbC, rzM3, Hr9P).
- Superior performance demonstrated through extensive experiments (Reviewers 2PbC, XLzJ, Hr9P).
- Clear paper presentation (all reviewers gave a score of 3 for clarity).

We would like to address the concerns raised by Reviewer rzM3, the only reviewer to provide a negative rating. Following our rebuttal, Reviewer rzM3 acknowledged that "most concerns" had been addressed and raised additional technical questions, which we answered. Although no further response was received from this reviewer, we believe all the key concerns have been effectively resolved in the discussion period:

- We understand that the novelty of our work is Reviewer rzM3's primary concern, while other concerns are mainly about clarifying the technical details. In our view, Reviewer rzM3's concerns regarding novelty merely outline high-level connections between Moscat and existing works in various fields, while overlooking our substantial innovations in methodology, theory, and analysis. We provided a detailed explanation in our rebuttal to address this concern specifically. All the other reviewers acknowledged our work's novelty (all the reviewers gave a score of 3 for originality). For example, in our discussion with Reviewer XLzJ, we reached the consensus that existing works lack the in-depth analysis needed to establish the connection between network depth and heterophily. In contrast, our work provides this crucial analysis.

- Reviewer rzM3 pointed out valid technical issues regarding the low labeling rate scenario, the homophilous graph scenario, and the homophily ratio estimation. In response, we conducted further experiments to address these issues. The results show that Moscat remains robust in both low labeling rate and homophilous graph scenarios. Moreover, our homophily ratio estimation achieves over 90% agreement with the ground truth while maintaining comparable performance on downstream tasks.

---

### Decision · Program_Chairs · 2025-09-17

**Decision:**

Accept (poster)

**Comment:**

This paper presents a compelling study on the generalization disparities of deeper GNNs across nodes with different homophily levels. The proposed method, Moscat, is a novel and effective post-training mixture-of-experts framework that leverages these insights to significantly improve performance. The theoretical analysis is sound, and the experimental validation is extensive and rigorous, demonstrating strong improvements across various architectures and datasets. While one reviewer had concerns about novelty, the majority recognized the work's contributions, and the authors have thoroughly addressed all technical questions in the rebuttal. The paper is well-motivated, clearly presented, and offers valuable insights for the community. I recommend acceptance.